# CRISPR/Cas9 mediated deletion of the adenosine A2A receptor enhances CAR T cell efficacy

Lauren Giuffrida[1,2,7], Kevin Sek [1,2,7], Melissa A. Henderson[1,2], Junyun Lai[1,2], Amanda X. Y. Chen[1,2], Deborah Meyran[1,2], Kirsten L. Todd[1,2], Emma V. Petley[1,2], Sherly Mardiana [1,2], Christina Mølck[3], Gregory D. Stewart[4], Benjamin J. Solomon[2], Ian A. Parish [1,2], Paul J. Neeson [1,2], Simon J. Harrison [2,5], Lev M. Kats [2,3], Imran G. House [1,2,8✉], Phillip K. Darcy [1,2,3,6,8✉] & Paul A. Beavis [1,2,8✉]

Adenosine is an immunosuppressive factor that limits anti-tumor immunity through the suppression of multiple immune subsets including T cells via activation of the adenosine $A_{2A}$ receptor ($A_{2A}R$). Using both murine and human chimeric antigen receptor (CAR) T cells, here we show that targeting $A_{2A}R$ with a clinically relevant CRISPR/Cas9 strategy significantly enhances their in vivo efficacy, leading to improved survival of mice. Effects evoked by CRISPR/Cas9 mediated gene deletion of $A_{2A}R$ are superior to shRNA mediated knockdown or pharmacological blockade of $A_{2A}R$. Mechanistically, human $A_{2A}R$-edited CAR T cells are significantly resistant to adenosine-mediated transcriptional changes, resulting in enhanced production of cytokines including IFNγ and TNF, and increased expression of JAK-STAT signaling pathway associated genes. $A_{2A}R$ deficient CAR T cells are well tolerated and do not induce overt pathologies in mice, supporting the use of CRISPR/Cas9 to target $A_{2A}R$ for the improvement of CAR T cell function in the clinic.

[1] Cancer Immunology Program, Peter MacCallum Cancer Centre, Melbourne, Vic, Australia. [2] Sir Peter MacCallum Department of Oncology, The University of Melbourne, Parkville, Vic, Australia. [3] Department of Pathology, University of Melbourne, Parkville, Vic, Australia. [4] Drug Discovery Biology, Monash Institute of Pharmaceutical Sciences, Monash University, Parkville, Vic, Australia. [5] Clinical Haematology and Centre of Excellence for Cellular Immunotherapies, Peter MacCallum Cancer Centre and Royal Melbourne Hospital, Melbourne, Vic, Australia. [6] Department of Immunology, Monash University, Clayton, Vic, Australia. [7] These authors contributed equally: Lauren Giuffrida, Kevin Sek. [8] These authors jointly supervised this work: Imran G. House, Phillip K. Darcy, Paul A. Beavis. ✉email: imran.house@petermac.org; phil.darcy@petermac.org; paul.beavis@petermac.org

Chimeric antigen receptor (CAR) T-cell therapy involves the ex vivo transduction of a patient's T cells with a CAR that recognizes a defined tumor antigen through a small-chain variable fragment, linked to intracellular signaling domains, most conventionally CD3ζ and either 4-1BB or CD28[1,2]. CAR T cells targeting the CD19 antigen are now FDA-approved for the treatment of relapsed B-cell acute lympho-blastic leukemia and aggressive lymphoma following remarkable clinical response rates and defined curative potential[3]. However, these effects have not been recapitulated in the solid tumor setting, where CAR T cells are faced with additional barriers such as tumor antigen heterogeneity, the requirement to traffic to the tumor site, and an immunosuppressive, hypoxic tumor microenvironment[1,2]. A major mediator of immunosuppression within the hypoxic tumor microenvironment is adenosine, a metabolite that is generated by the ectoenzymes CD73, CD39, and CD38[4,5]. Formative studies by the group of Sitkovsky identified that the extracellular adenosine-A2AR-cAMP axis is critically important in controlling immune responses, inflammatory tissue damage, and antitumor immunity[6,7]. Adenosine has four known G-protein-coupled receptors, $A_1R$, $A_{2A}R$, $A_{2B}R$, and $A_3R$, and while the activation of both $A_{2B}R$ and $A_{2A}R$ has been shown to elicit immunosuppressive effects, the suppression of T-cell activity is predominantly mediated by $A_{2A}R$, which leads to activation of adenylate cyclase and enhanced levels of intracellular cAMP[8]. $A_{2A}R$- deficient mice elicit significantly enhanced T-cell responses and notably this $A_{2A}R$-cAMP axis is nonredundant, since other cAMP-elevating G-protein-coupled receptors do not compensate for the loss of $A_{2A}R$[6]. These seminal findings have subsequently been confirmed by multiple laboratories, highlighting that targeting of $A_{2A}R$ can enhance T-cell-mediated antitumor immune responses[9–11] and blockade of $A_{2A}R$ with several small-molecule antagonists is now being tested in clinical trials for the treatment of a range of solid cancers[12]. Furthermore, this work has set the scene for alternative strategies to target this pathway, including therapeutics directed toward the upstream ectoenzymes CD73, CD39, and CD38, or the upstream Hypoxia-HIF-1α axis itself[8,10] signifying the clinical interest in targeting this pathway[13].

Given that $A_{2A}R$ activation potently suppresses T-cell function, a natural extension was to investigate the potential of targeting this pathway to enhance adoptive T-cell therapy, including CAR T-cell therapy. Previous work has demonstrated that transfection of $A_{2A}R/A_{2B}R$-targeting siRNA into anti-CMS4 CD8+ T cells enhanced their antitumor properties, thus presenting $A_{2A}R$ as an attractive target and confirmed that the immunosuppressive effects of $A_{2A}R$ are at least partially T-cell intrinsic[7]. Multiple groups have subsequently shown that targeting $A_{2A}R$ with small-molecule antagonists can enhance therapeutic responses evoked by conventional TCR-based adoptive cellular therapy[5,14,15], while our previous work indicates that targeting $A_{2A}R$ also enhances CAR T-cell responses in a solid tumor setting[16].

A key observation from this study was that while pharmacological antagonism of $A_{2A}R$ led to enhanced CAR T-cell function, the effect was more striking when CAR T cells were generated from $A_{2A}R^{-/-}$ donor mice with complete loss of receptor function. This led to the hypothesis that devising a gene-editing strategy to silence $A_{2A}R$ signaling (as opposed to partial and/or temporary inhibition mediated by pharmacological antagonists) would be advantageous in the context of CAR T-cell therapy. We therefore sought to investigate whether the use of CRISPR/Cas9 or shRNA-mediated targeting of $A_{2A}R$ could enhance CAR T-cell function. These technologies are well suited for combination with CAR T-cell production, since gene-editing procedures can be introduced in parallel with CAR transduction. In particular,

CRISPR/Cas9-mediated deletion of PD-1, TGFβ receptor II, or the phosphatase PTPN2 has previously been shown to enhance CAR T-cell function, leading to enhanced T-cell proliferation and production of proinflammatory cytokines[17–19].

In this work, we show that targeting $A_{2A}R$ by shRNA knock-down promotes the effector functions of murine CAR T cells following stimulation through the CAR and enhances CAR T-cell effector function in vivo, but is also associated with reduced persistence. In contrast, CRISPR/Cas9-mediated deletion of $A_{2A}R$ in both murine and human-derived CAR T cells abrogates the immunosuppressive effects of adenosine and enhances effector function, while having no deleterious effect on memory phenotype or persistence of CAR T cells. Furthermore, enhanced in vivo antitumor efficacy evoked by human $A_{2A}R$-edited CAR T cells is not associated with toxicity as defined by enzymatic readouts of hepatotoxicity and analysis of tissue sections. These results suggest that eliciting total knockout of $A_{2A}R$ using CRISPR/Cas9 is a superior therapeutic approach to enhance CAR T-cell function than either shRNA-mediated knockdown or combination with pharmacological antagonists of the receptor. This approach is readily translatable to the clinic given CRISPR/Cas9-edited CAR T cells are being used in ongoing trials using the same methodology as described herein[20]. Furthermore, targeting $A_{2A}R$ via CRISPR/Cas9-mediated editing is applicable to CAR T-cell therapy in multiple tumor types where adenosine signaling has been shown to suppress anti-tumor immunity, including breast cancer, ovarian cancer, lung cancer, acute myeloid leukemia, multiple myeloma, and non-Hodgkin's lymphoma[21–25].

## Results

**$A_{2A}R$ expression limits CAR T-cell effector function in a gene dose-dependent manner.** Our previous studies indicate that $A_{2A}R$ activation potently suppresses T-cell effector function and anti-tumor activity. To analyze the genome-wide impact of $A_{2A}R$ activation on CAR T cells specifically, we generated murine CAR T cells targeting human Her2 using retroviral transduction as per our previous work[16,26–28]. CAR T cells were then stimulated with an anti-CAR antibody in the presence or absence of NECA, an adenosine mimetic, as well as SCH58261, a selective $A_{2A}R$ antagonist. Analysis by 3′RNA-Seq revealed expected upregulation of genes associated with T-cell effector function following CAR activation such as *Ifng*, *Tnf*, and *Gzmb* (Fig. 1A). Importantly, in the context of cells activated through the CAR, transcriptional changes induced by NECA were fully reversible with addition of the $A_{2A}R$ antagonist SCH58261, demonstrating that adenosine acts on CAR T-cell effector function principally through $A_{2A}R$ activation (Fig. 1B). To assess the gene-dosage relationship of NECA-mediated $A_{2A}R$ activation, we compared anti-Her2 CAR T cells generated on $A_{2A}R^{+/+}$, $A_{2A}R^{+/-}$, and $A_{2A}R^{-/-}$ backgrounds. Transduction resulted in similar expression of the CAR and CD8:CD4 ratios in each of the three genotypes (Fig. 1C, D). These CAR T cells were then cocultured with E0771 (breast cancer) or 24JK (sarcoma) tumor cells engineered to express Her2, in the presence or absence of NECA. A major immunosuppressive phenotype mediated by $A_{2A}R$ activation is impaired cytokine production by T cells, while cytotoxic activity is less affected[7,16,29–31]. As expected, while NECA had no significant effect on direct CAR T-cell cytotoxic activity (Supplementary Fig. 1A), it significantly suppressed IFNγ and TNF production by wild-type $A_{2A}R^{+/+}$ but not $A_{2A}R^{-/-}$ CAR T cells following coculture with either E0771-Her2 (Fig. 1E, F) or 24JK-Her2 tumor cells (Supplementary Fig. 1B, C). The role of these cytokines in the indirect killing of tumor cells has previously been shown[32–34], and notably the addition of NECA led to a significant reduction in the cytokine-mediated killing activity of

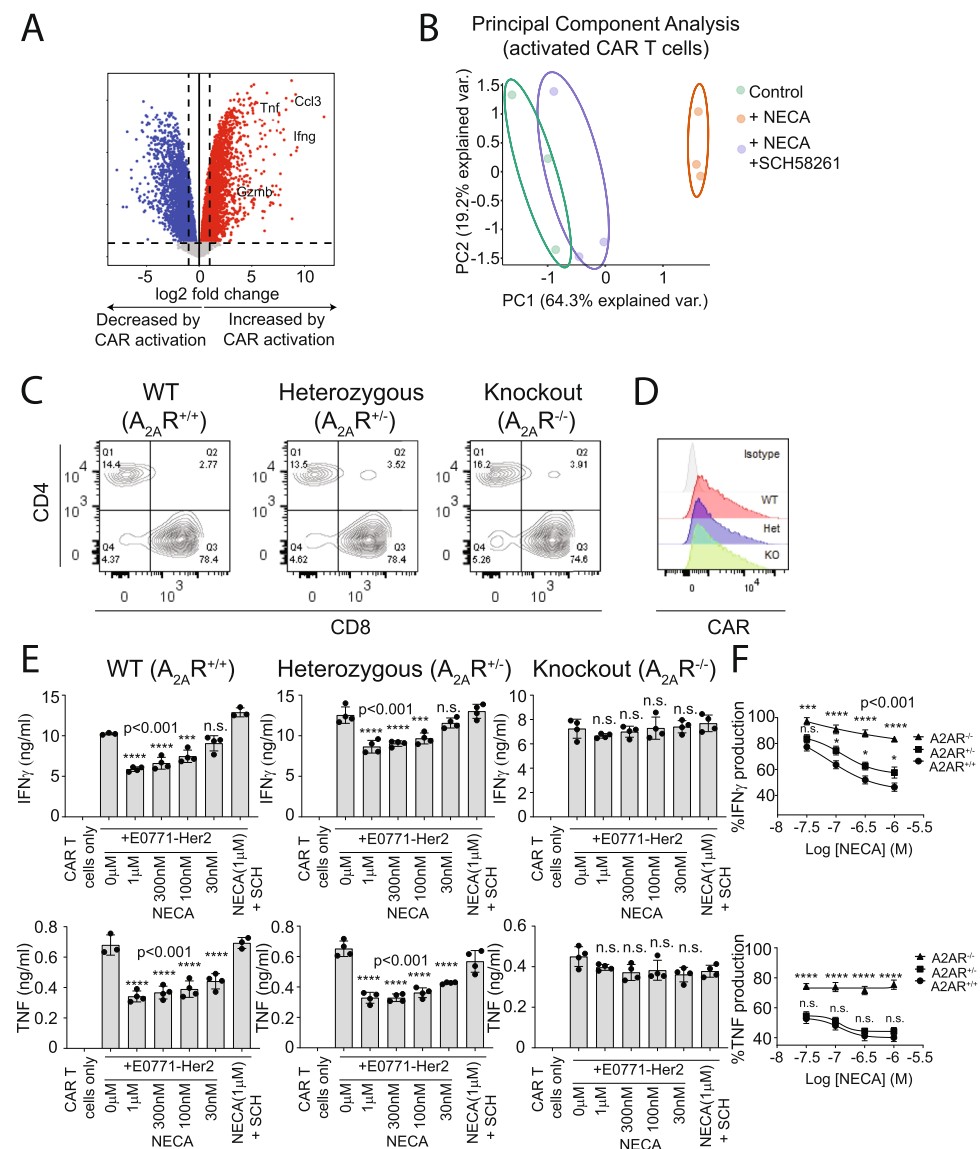

**Fig. 1 A$_{2A}$R stimulation significantly modulates the CAR T-cell transcriptome in a partially gene-dose-dependent manner. A, B** Anti-Her2 CAR T cells were generated from WT mice and after 7 days of culture in IL-2 (100 U/ml) and IL-7 (200 pg/ml) stimulated with plate-bound anti-myc Tag (anti-CAR) antibody in the presence or absence of NECA (1 μM) or SCH58261 (1 μM) for 8 h. Cells were then analyzed by 3'RNA-Seq. **A** Differential gene expression analysis, color indicates genes that are significantly upregulated (red) or downregulated (blue) following activation through the CAR. Statistical tests were performed with indicated R packages as outlined in "Methods". **B** Principal component analysis based on top 100 most variably expressed genes. **C–E** Anti-Her2 CAR T cells were generated from A$_{2A}$R$^{+/+}$ (wild type; WT), A$_{2A}$R$^{+/-}$, or A$_{2A}$R$^{-/-}$ mice as per (**A**). **C, D** Expression of (**C**) CD8, CD4, and (**D**) anti-Her2 CAR. **E, F** 1 × 10$^5$ anti-Her2 CAR T cells were cocultured with E0771-Her2 for 16 h in the presence or absence of SCH58261 (1 μM) and/or indicated concentrations of NECA. **E** Data represent the mean ± SD of triplicate/ quadruplicate cultures from a representative experiment of $n = 3$. Statistics indicate significance relative to 0 μM NECA condition. **F** Percentage production of IFNγ and TNF relative to the control condition for each genotype. The results were calculated from pooled data of $n = 3$ experiments presented as mean ± SEM. Nonlinear regression analysis. **E, F** Statistics indicate significance relative to wild-type values as calculated by a one-way ANOVA and Tukey's post hoc test (adjusted for multiple comparisons). Source data are provided as a Source Data file.

supernatants derived from these cocultures (Supplementary Fig. 1D, E). Heterozygous A$_{2A}$R$^{+/-}$ CAR T cells were partially resistant to NECA-mediated suppression in terms of their production of IFNγ, but this resistance was modest compared to A$_{2A}$R$^{-/-}$ CAR T cells. In addition, the suppression of TNF production by NECA was equivalent in A$_{2A}$R$^{+/+}$ and A$_{2A}$R$^{+/-}$ CAR T cells. Taken together, these data suggest significant redundancy in the levels of A$_{2A}$R expressed by CAR T cells, implying that knockdown approaches would need to achieve a greater than 50% reduction in the levels of A$_{2A}$R expression to be effective.

**shRNA-mediated A$_{2A}$R knockdown enhances CAR T-cell effector function**. To evaluate this further, we generated retroviruses expressing four independent shRNAs targeting A$_{2A}$R. These retroviruses also encoded the expression of truncated nerve growth factor receptor (NGFR), a commonly used marker gene, which was used for the purpose of isolating shRNA-transduced CAR T cells. Due to the lack of A$_{2A}$R antibodies suitable for flow cytometry, the efficiency of A$_{2A}$R knockdown was functionally assessed via measurement of cAMP production following stimulation with NECA, since activation of A$_{2A}$R results in accumulation of cAMP, the signaling molecule responsible for

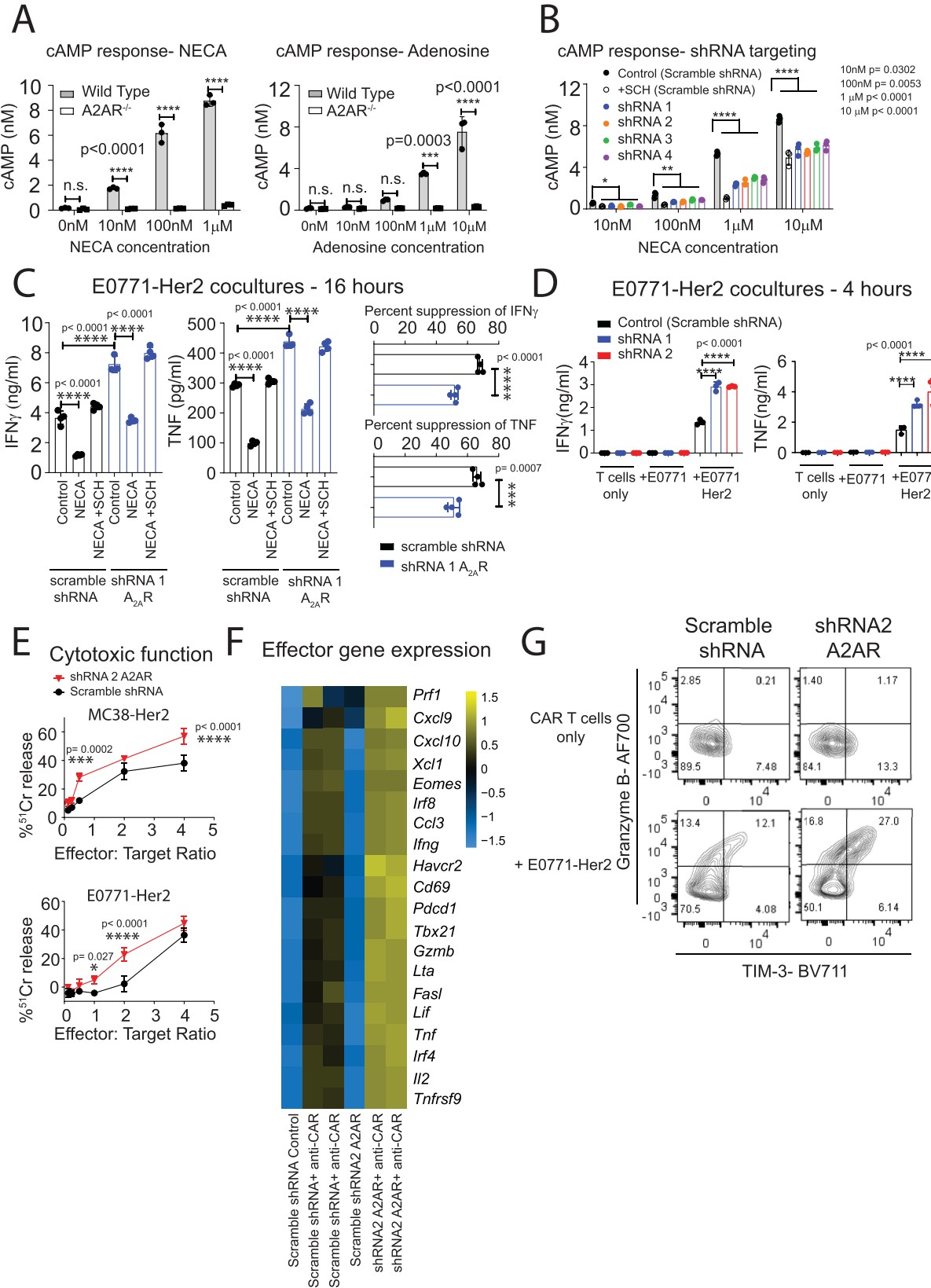

A2AR-mediated suppression[35]. To validate this assay, the cAMP response of wild-type or A2AR$^{-/-}$ CAR T cells to either NECA or adenosine was determined. As expected, wild-type CAR T cells elicited significantly stronger cAMP signaling when treated with either A2AR agonist (Fig. 2A). Moreover, the addition of the A2AR antagonist SCH58261 significantly reduced NECA-mediated

induction of cAMP, further confirming that this effect was A2AR-mediated (Fig. 2B). To assess the efficiency of knockdown in shRNA-transduced cells, and to assess the effects on CAR T-cell function, NGFR$^+$ cells were purified following transduction and utilized in downstream functional assays. All four A2AR-targeting shRNAs significantly suppressed NECA-mediated

**Fig. 2 $A_2AR$ knockdown enhances CAR T-cell effector function.** Anti-Her2 CAR T cells were generated as per Fig. 1 and, where indicated, cotransduced with shRNAs directed toward either $A_2AR$ or a nontargeting (scramble) shRNA control. NGFR$^+$ CAR T cells were isolated through MACS isolation prior to downstream assays. **A**, **B** cAMP accumulation following stimulation of CAR T cells with indicated doses of NECA or adenosine. Where indicated, SCH58261 (SCH, 1 μM) was added as a control. Representative experiments of $n = 2$. **B** Data represented as mean ± SD of triplicate replicates. Statistical tests indicated for scramble shRNA vs. other groups, one-way ANOVA. **C**, **D** $1 \times 10^5$ CAR T cells were cultured with $1 \times 10^5$ E0771-Her2 cells for **C** 16 h or (**D**) 4 h and supernatants analyzed by CBA to detect IFNγ and TNF. Data are presented as the mean ± SD of triplicate/quadruplicate cultures from a representative experiment of $n = 3$. **E** Anti-Her2 CAR T cells were cocultured with $1 \times 10^4$ of indicated $^{51}$chromium-labeled tumor cells at the indicated effector:target ratios for 4 hours. Chromium release was detected by an automated gamma counter (Wallac Wizard 1470). Data are shown as the mean ± SD of triplicate cultures. Statistics: two-way ANOVA. **A**–**E** ****$p < 0.0001$, ***$p < 0.01$, *<$p < 0.01$ *$p < 0.05$. **F** Heatmap for expression of indicated genes determined by NanoString analysis. **G** CAR T cells were transduced as per Fig. 1, but at day 2 post transduction, maintained in IL-7 and IL-15 (10 ng/ml). $2 \times 10^5$ CAR T cells were then cultured with $1 \times 10^5$ E0771-Her2 tumor cells for 72 h with $1 \times 10^5$ tumor cells replaced every 24 h. CAR T cells were then analyzed for their expression of TIM-3 and Granzyme B. Data shown represent concatenated triplicated samples from a representative experiment of $n = 2$. Source data are provided as a Source Data file.

cAMP production in CAR T cells relative to a scrambled shRNA control, leading to suppression of cAMP signaling by approximately 30–50% (Fig. 2B). In addition, we further validated $A_2AR$ knockdown at the mRNA level for the two most effective shRNAs through quantitative reverse transcription-polymerase chain reaction (Supplementary Fig. 2A). Having confirmed effective knockdown, we next assessed the impact of $A_2AR$ knockdown on CAR T-cell effector function upon coculture with Her2$^+$ tumor cells. For functional analyses, we performed studies using the two hairpins that mediated the most robust knockdown of $A_2AR$. Transduction of CAR T cells with these shRNAs resulted in equivalent expression of the CAR between groups (Supplementary Fig. 2B) and positive selection of NGFR$^+$ cells led to a highly purified (~90%) population (Supplementary Fig. 2C). Despite effective suppression of $A_2AR$ signaling (as assessed by reduced cAMP levels), shRNA-mediated knockdown of $A_2AR$ had only a moderate effect in reducing NECA-mediated suppression of IFNγ and TNF production (Fig. 2C). This suggests that partial $A_2AR$ function is sufficient to elicit the immunosuppressive effects mediated by adenosine. Nevertheless, in the absence of NECA stimulation, $A_2AR$ knockdown significantly enhanced both the cytokine production and, to a lesser extent, the cytotoxic function of CAR T cells against E0771-Her2 and MC38-Her2 (colon adenocarcinoma) tumor cells (Fig. 2C–E). More broadly, $A_2AR$ knockdown enhanced the transcriptional expression of a wide range of effector-related genes following stimulation through the CAR (Fig. 2F) and increased the emergence of an activated TIM-3 $^+$Granzyme B$^+$ subset upon serial coculture with Her2$^+$ tumor cells to mimic differentiation within the tumor microenvironment (Fig. 2G). This suggests that $A_2AR$ knockdown modulated CAR T-cell differentiation toward a more effector-like phenotype, potentially through reduction of residual $A_2AR$ signaling that exists in the absence of acute NECA stimulation. We next evaluated the capacity of shRNA-mediated $A_2AR$-knockdown CAR T cells to elicit enhanced antitumor activity in vivo. These studies were performed in a syngeneic human Her2 transgenic (Her2 Tg) mice strain that we have previously characterized and allows for the assessment of CAR T-cell function in an immunocompetent setting, including adenosine-producing immunosuppressive subsets[16,26,28,36]. Mice were injected with $2 \times 10^5$ E0771-Her2 tumor cells into the fourth mammary fat pad and, once tumors were established, the mice were preconditioned with 4-Gy irradiation and treated with $1 \times 10^7$ CAR T cells on two consecutive days. Knockdown of $A_2AR$ resulted in a modest, but significant enhancement of tumor growth inhibition relative to control CAR T cells (Fig. 3A; see schematic Supplementary Fig. 3A). Increased in vivo efficacy by $A_2AR$-knockdown CAR T cells was associated with increased expression of effector-related genes in tumor-infiltrating CD8$^+$NGFR$^+$ CAR T cells such as IFNγ, TNF, IRF4, and Granzyme B (Fig. 3B, C). Further analysis of the phenotype

of CD4$^+$NGFR$^+$ CAR T cells indicated that $A_2AR$ knockdown enhanced their expression of IFNγ and Ki-67, similarly to CD8$^+$ CAR T cells (Fig. 3D). Our data therefore clearly indicate that $A_2AR$ knockdown enhances CAR T-cell effector function. It was recently shown, however, that enhanced CAR T-cell effector function can be linked to reduced T-cell persistence[37] and so we assessed the persistence of CAR T cells following $A_2AR$ knockdown. Analysis of spleens revealed that $A_2AR$ knockdown significantly reduced the numbers of CAR T cells at day 9 post treatment (Fig. 3E). Taken together, these data suggest that while $A_2AR$ knockdown enhanced CAR T-cell effector function, it also reduced persistence, which likely restricted the extent of the therapeutic benefit observed.

**CRISPR/Cas9-mediated deletion of $A_2AR$ enhances CAR T-cell efficacy.** Our previous studies with $A_2AR^{-/-}$ CAR T cells or endogenous T cells in the context of tumor-bearing $A_2AR$-knockout mice indicated that complete loss of $A_2AR$ signaling had no adverse effects on T-cell persistence and memory[16,31]. Moreover, a recent study showed that $A_2AR$ deletion enhanced T-cell persistence in the context of LCMV infection[38]. Given the discrepancies between these observations and our present results using shRNA-mediated knockdown of $A_2AR$, we next investigated whether targeting $A_2AR$ using a CRISPR/Cas9 methodology, which results in complete loss of $A_2AR$ function in edited cells, could enhance CAR T-cell effector function without adversely affecting T-cell persistence. For CRISPR/Cas9-mediated deletion of $A_2AR$, recombinant Cas9 and a single-guide RNA (sgRNA) targeting $A_2AR$ (or nontargeting control) were delivered to naive splenocytes as a ribonucleoprotein (RNP) via electroporation prior to our conventional T-cell activation and transduction protocol. Importantly, our methodology utilizing electroporation-mediated delivery of RNP to T cells is similar to that implemented in recent clinical trials[20,39], and is amenable to CAR T-cell manufacturing protocols. To validate this approach in our hands, we first applied this methodology utilizing a guide targeting Thy1, a protein expressed ubiquitously on T cells. Using this approach, we were able to identify a complete loss of Thy1 expression in ~75% of CAR T cells with a similar efficiency observed in CD8$^+$ and CD4$^+$ CAR T cells (Supplementary Fig. 4A). Having validated this methodology, we next applied this to targeting $A_2AR$. Delivery of the $A_2AR$-targeting RNP affected neither the ratio of CD8:CD4 CAR T cells, nor the efficiency of CAR transduction (Supplementary Fig. 4B, C). Successful deletion of $A_2AR$ was confirmed through sequencing of genomic DNA and subsequent analysis via the Interference of CRISPR Edits (ICE) platform, which revealed that knockout was highly efficient (87.3% ± 5.7% SEM) (Fig. 4A). To assess the functional consequence of $A_2AR$ editing, we next investigated cAMP signaling in response to NECA stimulation of CAR T cells. In

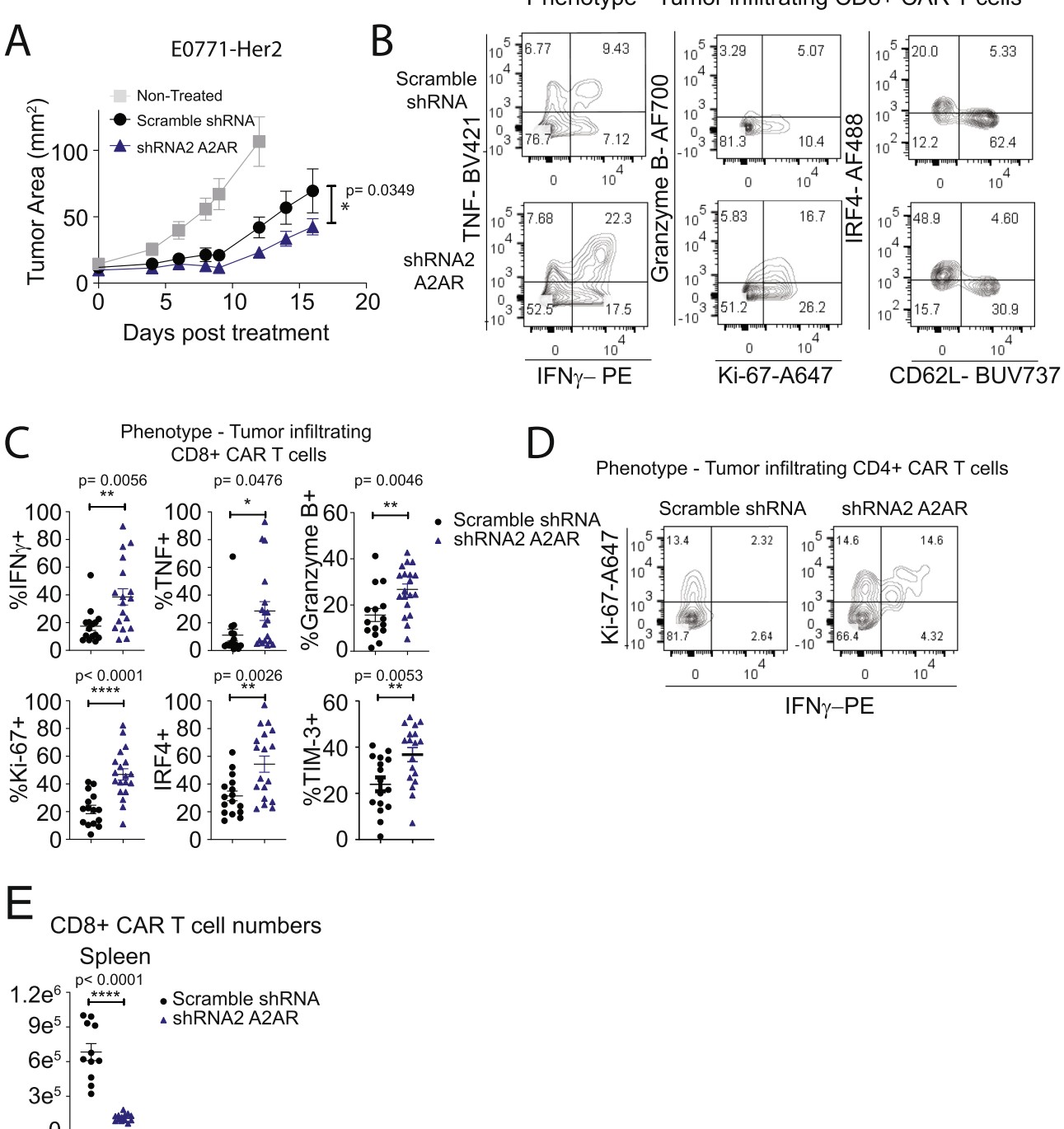

**Fig. 3 A₂ₐR knockdown promotes an effector-like CAR T-cell phenotype in vivo.** C57BL/6 Her2 mice were injected with $2 \times 10^5$ E0771-Her2 tumor cells into the fourth mammary fat pad and treated with anti-Her2 CAR T cells transduced and cultured in IL-7 and IL-15 as per Fig. 2G. Once tumors were established, $1 \times 10^7$ CAR T cells were injected intravenously on two subsequent days following 4-Gy total body irradiation. Mice were treated with 50,000U of IL-2 on days 0–4 post CAR T-cell treatment. **A** Tumor growth shown as the mean ± SEM of $n = 5$ (CAR T-cell treated) or 6 (nontreated) mice per group from a representative experiment: $n = 2$, *$p < 0.05$, two-way ANOVA. **B–E** Spleens and tumors were excised 9 days post treatment and the number and phenotype of NGFR⁺ CAR T cells determined by flow cytometry. **B, D** Expression from concatenated samples ($n = 7$ per group) from a representative experiment. **C, E** Data represent the mean ± SEM of $n = 11$–19 mice⁺ from 2 (**E**) or 3 (**C**) replicate experiments (see ⁺ below for details). Each data point represents an individual mouse. *$p < 0.05$, **$p < 0.01$, ***$p < 0.001$, ****$p < 0.0001$, unpaired $t$ test. ⁺ IFNγ, TNF, TIM-3 control $n = 15$, shRNA A₂ₐR $n = 18$, Granzyme B and Ki-67, control $n = 15$, shRNA A₂ₐR $n = 19$, IRF4, control $n = 16$, shRNA A₂ₐR $n = 18$, numbers in spleen control $n = 11$, shRNA A₂ₐR $n = 13$. Data from some control samples are also presented in ref. [36]. Source data are provided as a Source Data file.

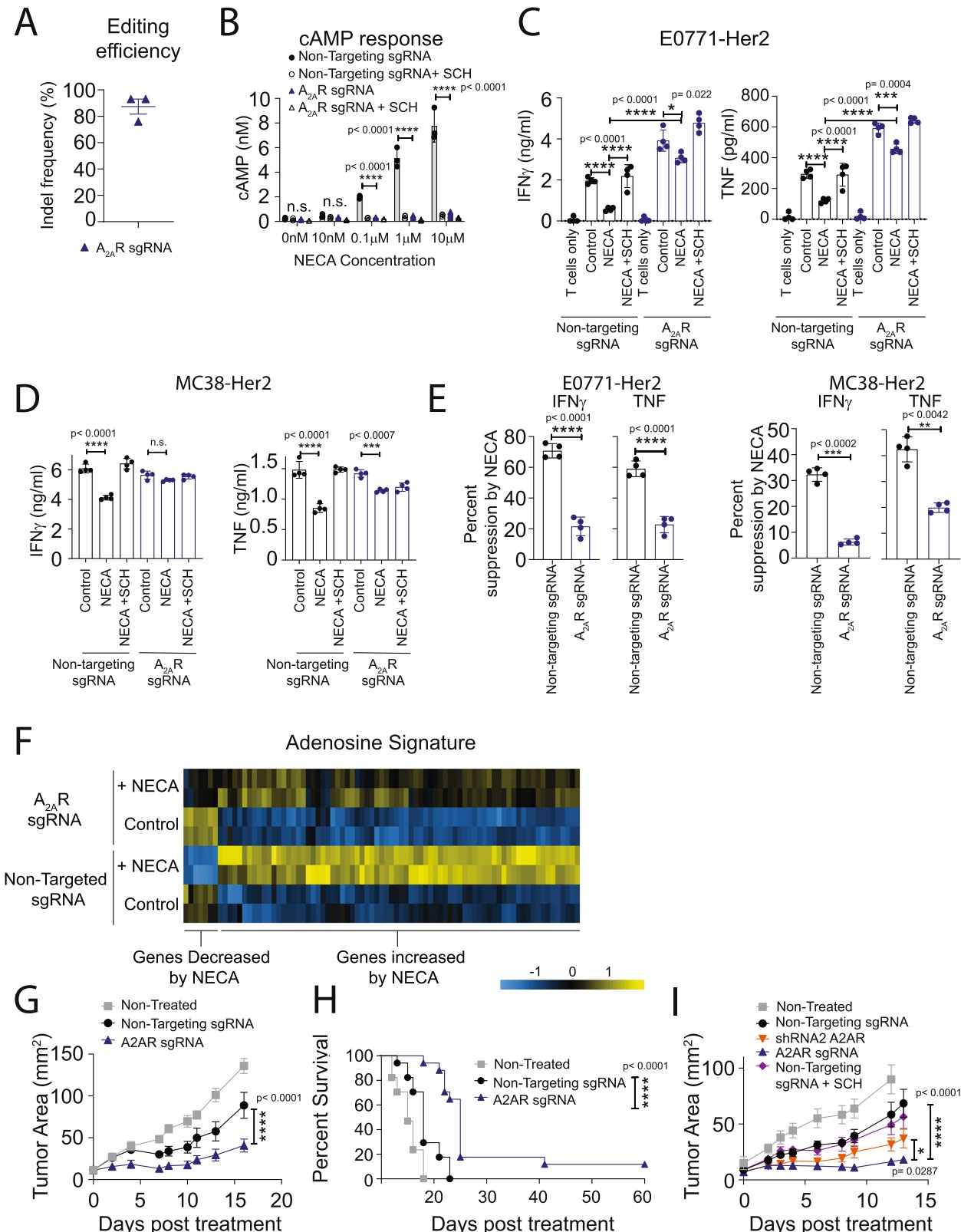

contrast to shRNA-mediated knockdown of $A_{2A}R$, CRISPR/Cas9-mediated knockout of $A_{2A}R$ almost completely attenuated cAMP signaling (Fig. 4B). In addition, $A_{2A}R$ targeting with CRISPR/Cas9 resulted in almost complete resistance to NECA-mediated suppression of IFNγ and TNF production upon coculture with Her2-positive tumor cells (Fig. 4C–E). However, CRISPR/Cas9-mediated knockout of $A_{2A}R$ did not enhance the cytotoxicity of

CAR T cells measured at either 4 or 16 h, consistent with our previous observations indicating that short-term $A_{2A}R$ activation modulates CAR T-cell cytokine production but not cytotoxic function in vitro (Supplementary Fig. 4D, E)[16]. Using genome-wide analysis by 3′RNA-Seq following stimulation through the CAR as in Fig. 1A, we investigated the effect of CRISPR/Cas9-mediated targeting of $A_{2A}R$ on NECA-mediated transcriptional

**Fig. 4 CRISPR/Cas9-mediated A$_{2A}$R deletion enhances CAR T-cell efficacy.** Anti-Her2 CAR T cells were generated as per Fig. 1 following CRISPR/Cas9-mediated editing of CAR T cells with either an A$_{2A}$R sgRNA or a nontargeting sgRNA control. **A** INDEL frequency as determined by ICE analysis. Data represent the mean ± SEM of three replicate experiments. **B** cAMP accumulation in response to NECA determined as per Fig. 2A. Data represented as the mean ± SD of triplicates. Representative experiment of $n = 2$. **C, D** $1 \times 10^5$ CAR T cells were cocultured with (**C**) $1 \times 10^5$ E0771-Her2 or (**D**) $1 \times 10^5$ MC38-Her2 tumor cells in the presence or absence of NECA (1 μM) or SCH58261 (1 μM) for 8 h and supernatants analyzed by CBA. Data represent the mean ± SD of quadruplicate cultures from a representative experiment of $n = 3$. *$p < 0.05$, ***$p < 0.001$, ****$p < 0.0001$, one-way ANOVA. **E** Data from **C, D** calculated as percentage suppression of IFNγ and TNF production mediated by NECA on control and A$_{2A}$R-edited CAR T cells. ****$p < 0.0001$ paired $t$ test, data presented as mean ± SD of quadruplicate cultures ($n = 4$) from a representative experiment of $n = 3$. **F** Anti-Her2 CAR T cells were stimulated with plate-bound anti-myc Tag (anti-CAR) antibody in the presence or absence of NECA (1 μM) for 8 h and then analyzed by 3′RNA-seq. Heat map of expression of murine adenosine signature-related genes is shown. **G–I** E0771-Her2 tumor-bearing mice were treated as per Fig. 3 with control, A$_{2A}$R knockdown, or A$_{2A}$R CRISPR/Cas9-edited CAR T cells. **G** Data represent the mean ± SEM of $n = 5$ (non-targeting sgRNA) or 6 (nontreated, A$_{2A}$R sgRNA) of a representative experiment of $n = 4$. **I** Data represent the mean ± SEM of $n = 6$ (nontreated, nontargeting sgRNA, A$_{2A}$R sgRNA), 5 (nontargeting sgRNA + shRNA 2), or 4 (nontargeting sgRNA + SCH58261) from a representative experiment of $n = 2$ (**I**). **G, I** Tumor growth ****$p < 0.0001$, *$p < 0.05$. Two-way ANOVA. **H** Survival defined as when tumors exceeded 100 mm$^2$, ****$p < 0.0001$ Log-Rank test. Data indicative of 17 mice per group combined from 3 experiments. Source data are provided as a Source Data file.

changes. Using an FDR < 0.05 cutoff, we identified the genes most highly regulated following NECA stimulation, which we designated an "adenosine signature" (Supplementary Data 1). As expected, the modulation of genes comprising this adenosine signature was significantly attenuated following A$_{2A}$R deletion (Fig. 4F; Supplementary Data 1). We next investigated the in vivo antitumor efficacy of A$_{2A}$R-edited CAR T cells in mice bearing E0771-Her2 tumors. Strikingly, CRISPR/Cas9-mediated deletion of A$_{2A}$R significantly enhanced the antitumor efficacy of transferred CAR T cells (Fig. 4G, H; see schematic Supplementary Fig. 3B). Notably, these therapeutic effects were more pronounced than those observed with either shRNA-mediated knockdown of A$_{2A}$R or with pharmacological-mediated A$_{2A}$R blockade (Fig. 4I, Supplementary Fig. 4F). Furthermore, mice previously cured following therapy with A$_{2A}$R-edited CAR T cells were resistant to a secondary challenge with E0771-Her2 on the opposite mammary fat pad, suggesting that A$_{2A}$R-edited CAR T cells were capable of evoking memory recall responses (Supplementary Fig. 4G).

**CRISPR/Cas9-mediated editing of A$_{2A}$R enhances both CD8$^+$ and CD4$^+$ CAR T-cell function.** As adenosine has been shown to modulate both CD8$^+$ and CD4$^+$ T-cell function, we next investigated the impact of A$_{2A}$R deletion on CD8$^+$ and CD4$^+$ CAR T-cell subsets and the subsequent impact on therapeutic response. RNA-Seq analysis of sorted CD8$^+$ and CD4$^+$ CAR T cells following activation through the CAR indicated that the *Adora2a* gene was similarly upregulated on CD8$^+$ and CD4$^+$ CAR T cells following activation through the CAR (Fig. 5A). Addition of NECA significantly modulated the transcriptional profile of both CD8$^+$ and CD4$^+$ CAR T cells, an effect that was completely ablated following CRISPR/Cas9-mediated deletion of A$_{2A}$R (Fig. 5B, C). Analysis of genes significantly (FDR ≤ 0.05, log$_2$ fold change ≥ 0.75 or ≤ −0.75) modulated by NECA in each cell type indicated that NECA significantly suppressed the expression of 323 genes in CD8$^+$ T cells and 309 genes in CD4$^+$ T cells, and of these, 117 genes were common to both CD8$^+$ and CD4$^+$ CAR T cells (Fig. 5D). Genes that were commonly suppressed in both CD8$^+$ and CD4$^+$ CAR T cells included a number of cytokines such as *Ifng*, *Tnf*, *Il2*, *Ccl3*, and *Xcl1*. Specifically within the CD4$^+$ CAR T-cell population, NECA suppressed the expression of a number of members of the TNF family (*Lta*, *Tnfsf14*) and genes associated with NFκB signaling (*Ikbke* and *Rel*) or signaling downstream of the TCR/CAR (*Nfatc1*, *Nfatc2*) (Fig. 5E). To assess the relative importance of A$_{2A}$R deletion in CD8$^+$ and CD4$^+$ CAR T cells for their in vivo efficacy, we purified CD8$^+$ CAR T cells from the bulk CAR T-cell product following CRISPR/Cas9 editing of the A$_{2A}$R receptor (Fig. 5F).

The enhanced therapeutic activity of A$_{2A}$R-edited CAR T cells was observed with either the bulk or CD8$^+$-enriched CAR T-cell product (Fig. 5G), indicating that the deletion of A$_{2A}$R from CD8$^+$ CAR T cells was sufficient to elicit an enhanced therapeutic response.

**CRISPR/Cas9-mediated deletion of A$_{2A}$R enhances CAR T-cell effector function with no adverse effect on peripheral T-cell persistence.** To investigate the mechanism underlying the enhanced therapeutic efficacy mediated by CRISPR/Cas9-mediated targeting of the A$_{2A}$R, we next analyzed the phenotype and number of CAR T cells in mice undergoing therapy. CRISPR/Cas9-mediated deletion of A$_{2A}$R did not affect CAR T-cell persistence in either the spleen or peripheral blood of treated mice (Fig. 6A, B). However, analysis of the phenotype of intratumoral CD8$^+$ CAR T cells revealed that CRISPR/Cas9-mediated deletion of the A$_{2A}$R enhanced the expression of a number of effector-related genes, including Granzyme B, Ki-67, and IRF4 and immune checkpoints PD-1 and TIM-3, which was commensurate with a reduction in CD62L expression. Taken together, these data are indicative of a transition to a more effector-like phenotype within tumors following CRISPR/Cas9-mediated deletion of A$_{2A}$R (Fig. 6C–E, gating strategy Supplementary Fig. 5A). Notably, a head-to-head comparison revealed that the number of CAR T cells recovered from the spleens at day 9 post treatment was significantly greater following CRISPR/Cas9-mediated deletion of A$_{2A}$R when compared to A$_{2A}$R-mediated knockdown via shRNA (Fig. 6F). Furthermore, shRNA-mediated knockdown of A$_{2A}$R was associated with reduced expression of CD62L on CD8$^+$ CAR T cells within the spleen, potentially explaining the reduced persistence relative to CRISPR/Cas9-edited CAR T cells (Fig. 6G).

**CRISPR/Cas9-mediated deletion of A$_{2A}$R enhances effector function of human CAR T cells.** Having established that CRISPR/Cas9-mediated A$_{2A}$R deletion enhanced the effector function of murine CAR T cells, we next investigated this in the context of human CAR T cells targeting a clinically relevant tumor antigen. To do this, we retrovirally transduced human CAR T cells with an anti-Lewis Y CAR currently being used in a phase Ib clinical trial[40,41] (NCT03851146), and then performed CRISPR/Cas9-mediated editing using either a nontargeting sgRNA or an A$_{2A}$R-targeting sgRNA. As observed in murine CAR T cells, CRISPR/Cas9-mediated deletion of A$_{2A}$R led to successful editing of more than 75% of *ADORA2A* gene alleles (Fig. 7A) and therefore potently attenuated the effect of NECA in inducing cAMP accumulation (Fig. 7B). A$_{2A}$R deletion did not affect the composition of the CAR T-cell product in terms of the

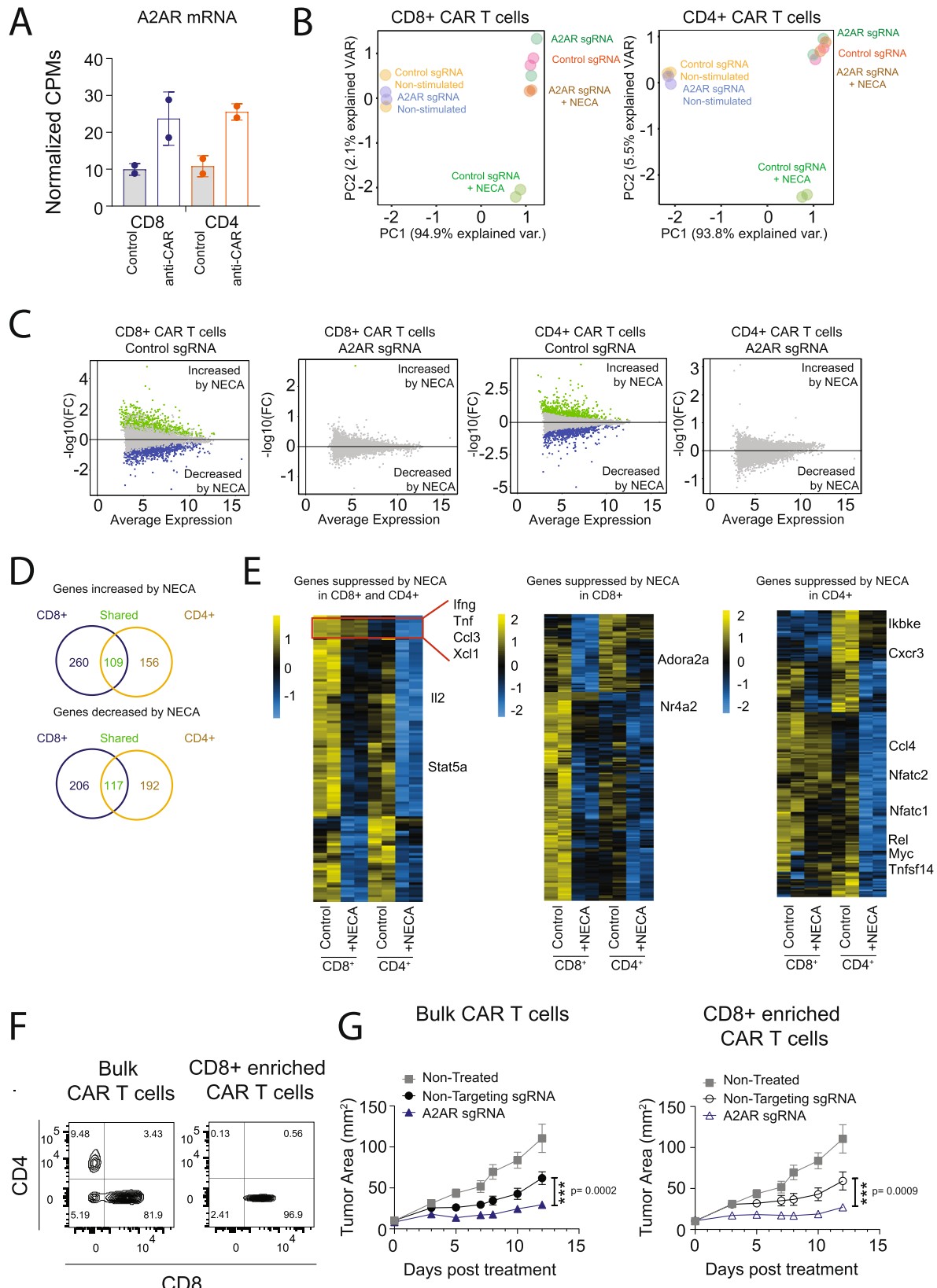

CD8:CD4 ratio (Supplementary Fig. 6A, B) nor the proportion of CD8[+] CAR T cells with a stem cell memory phenotype (CD45RA[+]CD62L[+]CD27[+]), which has previously been shown to associate with CAR T-cell persistence and clinical outcomes (Fig. 7C)[42]. Moreover, an analysis of potential off-target effects induced by this $A_{2A}R$-targeting sgRNA indicated no detectable

editing at the only three predicted off-target sites (Supplementary Fig. 7A–C). Similarly to previous studies using human CAR T cells targeting Her2, mesothelin, or CD19[16,43,44], the production of IFNγ and TNF by anti-Lewis Y CAR T cells was significantly suppressed by NECA during coculture with Lewis Y[+] OVCAR-3 tumor cells, an effect that was reversible through the

**Fig. 5 Targeting A₂ₐR by CRISPR/Cas9 enhances the effector function of both CD8⁺ and CD4⁺ CAR T cells.** Anti-Her2 CAR T cells were generated as per Fig. 4 following CRISPR/Cas9-mediated editing of CAR T cells with either an A₂ₐR sgRNA or a nontargeting sgRNA control. **A–E** $1.2 \times 10^6$ CAR T cells were stimulated with anti-myc Tag (anti-CAR) for 5 h in the presence or absence of NECA (1 μM) and cells then isolated via FACS sorting into CD4⁺ and CD8⁺ subsets. Significantly modulated genes were defined based on a threshold of log₂FC (fold change) of ≥0.75 or ≤−0.75 and FDR < 0.05 > log² 0.75-fold change and FDR < 0.05. Statistical tests were performed with indicated R packages as outlined in the "Methods". **A** CPMs for *Adora2a* gene in CD8⁺ and CD4⁺ CAR T cells before (control) and after (anti-CAR) activation. Data shown as mean ± SD of technical duplicates. **B** Principal component analysis based on top 100 most variable genes. **C** MA plot for genes increased or decreased by NECA in the indicated subsets. Significant genes are highlighted in green (increased by NECA) or blue (decreased by NECA). **D** Venn diagram showing the number of genes significantly modulated by NECA in each subset. **E** Heat maps for genes significantly suppressed by NECA in CD8⁺, CD4⁺, or both CD8⁺ and CD4⁺ CAR T cells. **F**, **G**. Anti-Her2 CAR T cells were generated as per Fig. 4G and CD8⁺ enrichment performed where indicated. Mice received one dose of $2 \times 10^7$ anti-Her2 CAR T cells. **F** Flow cytometry analysis of CD8 and CD4 expression within the CAR T-cell population before and after CD8⁺ enrichment. **G** Tumor growth represented as the mean ± SEM of six (nontreated, A₂ₐR sgRNA CD8⁺-enriched CAR T cells), seven (non-targeting sgRNA, bulk CAR T cells, and non-targeting sgRNA, CD8⁺-enriched CAR T cells), or eight (A₂ₐR sgRNA, bulk CAR T cells) mice per group, $n = 1$ experiment. ***$p < 0.001$. Two-way ANOVA. Source data are provided as a Source Data file.

addition of the A₂ₐR antagonist SCH58261 (Fig. 7D). We next assessed the effect of CRISPR/Cas9-mediated targeting of A₂ₐR on cytokine production of CAR T cells following coculture with Lewis Y⁺ tumor cells. CRISPR/Cas9-mediated editing of A₂ₐR rendered CAR T cells resistant to NECA in terms of their capacity to secrete IFNγ, TNF, and MIP1α (CCL3) upon coculture with either OVCAR-3 cells or MCF7 cells, another Lewis Y⁺ tumor cell line (Fig. 7E). With anti-Lewis Y CAR T cells generated across multiple donors, the deletion of A₂ₐR significantly reduced the suppressive effect of NECA on both IFNγ and TNF (Fig. 7F). By contrast, neither NECA nor A₂ₐR deletion significantly modulated the cytotoxic capacity of CAR T cells upon coculture with OVCAR-3 tumor cells in either 4-h (short-term) or 16-h (long-term) chromium release assays (Supplementary Fig. 6C–E). Notably, this is consistent with our previous observations that A₂ₐR signaling mediates the immune suppression of CAR T cells predominantly through impacting their cytokine production, rather than cytotoxic function in short-term assays[16]. To assess the effect of A₂ₐR stimulation at a genome-wide level, we next assessed the transcriptional profile of CAR T cells following A₂ₐR stimulation. In a setup analogous to our previous experiments with murine CAR T cells (Figs. 1A, B and 4E), we utilized an anti-idiotype antibody to stimulate the anti-Lewis Y CAR prior to 3′ RNA-seq analysis. As expected, stimulation of control or A₂ₐR-edited CAR T cells with the anti-idiotype antibody led to a significant upregulation of a number of effector-related genes. Principal component analysis revealed that the addition of NECA significantly modulated the phenotype of control CAR T cells but had minimal impact on the transcriptional profile of A₂ₐR-edited anti-Lewis Y CAR T cells (Fig. 7G). Through analysis of all genes significantly regulated by NECA in control CAR T cells (FDR < 0.05, fold change >2 or < −2), we generated an "adenosine signature" for human CAR T cells. Analysis of the expression of these genes indicated that A₂ₐR-edited CAR T cells had an attenuated response to NECA stimulation (Supplementary Fig. 6F, Supplementary Data 2). Unbiased analysis of the transcriptome of A₂ₐR-edited CAR T cells relative to control CAR T cells following CAR activation in the presence of NECA revealed a significant enrichment of several KEGG gene sets and, in particular, pathways pertaining to cytokine and JAK/STAT signaling (Supplementary Fig. 6G). These pathways were significantly downregulated in control- but not A₂ₐR-edited anti-Lewis Y CAR T cells following the addition of NECA (Fig. 7G–I). These gene lists include several pro-inflammatory cytokines and *IL2RA*, which encodes CD25, the receptor for IL-2, that were sensitive to NECA-mediated suppression, and were therefore significantly downregulated in control- but not A₂ₐR-edited CAR T cells following the addition of NECA (Supplementary Fig. 6H; Supplementary Data 2).

**CRISPR/Cas9-mediated A₂ₐR deletion enhances the in vivo efficacy of human CAR T cells.** We next investigated the capacity of CRISPR/Cas9 A₂ₐR-mediated editing of anti-Lewis Y CAR T cells to enhance their antitumor efficacy in vivo. To examine this, we treated NSG mice bearing OVCAR-3 tumors, which express high levels of CD73 (Supplementary Fig. 8A), with A₂ₐR-edited or control anti-Lewis Y CAR T cells. Strikingly, A₂ₐR deletion significantly enhanced CAR T-cell-mediated inhibition of tumor growth (Fig. 8A, see schematic Supplementary Fig. 3C), leading to significantly prolonged survival of mice (Fig. 8B), a finding that was replicated using CAR T cells generated from three independent donors (Supplementary Fig. 8B). Notably, A₂ₐR editing led to increased numbers of both CD8⁺ and CD4⁺ CAR T cells in the peripheral blood at day 8 post treatment relative to control CAR T cells (Fig. 8C), and did not overtly affect the memory phenotype of CD8⁺ (Fig. 8D) or CD4⁺ (Supplementary Fig. 8C) CAR T cells. Analysis of intratumoral CAR T cells revealed that CRISPR/Cas9-mediated editing of A₂ₐR significantly enhanced the proportion of both CD8⁺ and CD4⁺ CAR T cells expressing IFNγ and TNF, while the overall numbers of CAR T cells were unaffected (Fig. 8E, F, Supplementary Fig. 8D, gating strategy Supplementary Fig. 5B). To ascertain the safety of this approach, we assessed enzymatic readouts of renal and hepatotoxicity within the serum of mice undergoing therapy (Fig. 8G) and took sections of lungs, liver, and spleen at the experimental endpoint (Fig. 8H). Importantly, neither control nor A₂ₐR-edited CAR T cells modulated the concentration of alanine transaminase (ALT), aspartate aminotransferase (AST), or urea nor induced significant pathology as determined by hematoxylin and eosin staining. Taken together, these results indicate that targeting A₂ₐR using CRISPR/Cas9 methodology significantly improves the efficacy of both murine and human CAR T cells and is superior to shRNA-mediated knockdown or pharmacological A₂ₐR blockade.

## Discussion

Adenosine-mediated immunosuppression is now established as a major immune checkpoint for tumor-infiltrating T cells, limiting the efficacy of both conventional chemotherapy and immunotherapy approaches[4,8,13]. The ectoenzymes responsible for adenosine production, CD73 and CD39, are overexpressed in several cancer types, and in many cases, this is associated with poor prognosis[21,45–49]. A number of strategies have been developed to target this axis, with both antibodies to CD73, CD39, and CD38 and small-molecule antagonists of A₂ₐR and A₂ᵦR in clinical development[13,50,51]. Given that T cells express high levels of A₂ₐR, and that this receptor is responsible for the suppressive effect of adenosine on T cells, there has been significant interest in utilizing small-molecule A₂ₐR antagonists to enhance the efficacy

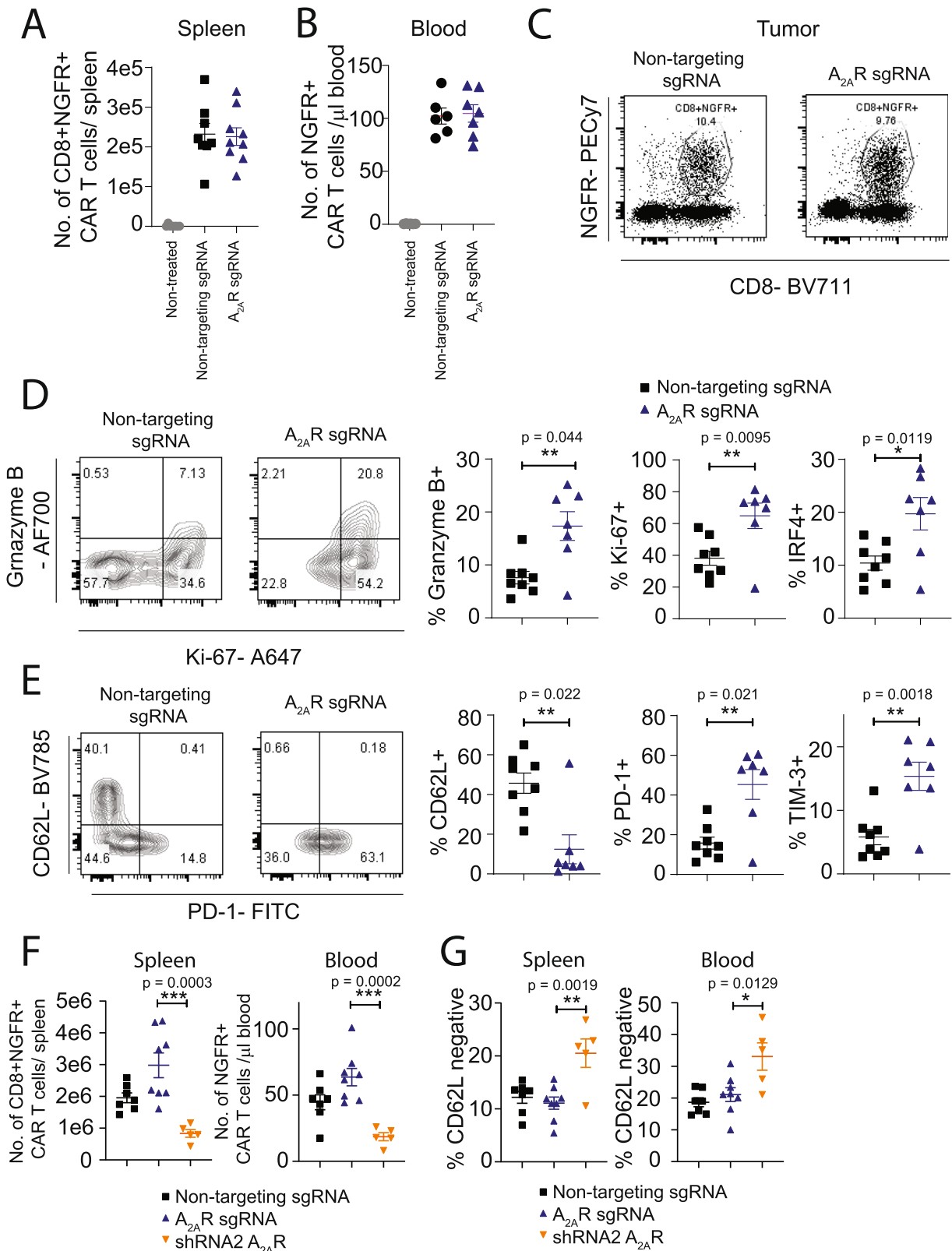

of ACT and CAR T cells. A number of small-molecule antagonists targeting $A_{2A}R$ have been shown to enhance the capacity of adoptively transferred T cells to elicit enhanced antitumor immunity, including SCH58261[15,43], KW6002[14], and CPI-444[5]. Our previous studies using murine CAR T cells revealed that while $A_{2A}R$ blockade with SCH58261 enhanced the in vivo efficacy of CAR T cells, the effects were more marked when CAR

T cells derived from $A_{2A}R$-deficient mice were used[16]. We therefore hypothesized that gene-targeting approaches designed to knock down/knock out $A_{2A}R$ would lead to significantly improved efficacy and selectivity relative to pharmacological $A_{2A}R$ blockade. To date, limited studies have been performed using genetic-based approaches to target adenosine signaling in CAR T cells. Two previous studies, including our own, have

**Fig. 6 CRISPR/Cas9-mediated $A_{2A}R$ deletion promotes CAR T-cell effector function without impacting CAR T-cell persistence.** E0771-Her2 tumor-bearing mice were treated with anti-Her2 CAR T cells, generated following CRISPR/Cas9-mediated editing as per Fig. 4. **A** The number of $CD8^+NGFR^+$ CAR T cells recovered from spleens of treated mice at day 7 post treatment. $n = 8$ (nontreated, nontargeting sgRNA) or $n = 9$ ($A_{2A}R$ sgRNA). **B** The number of $NGFR^+$ CAR T cells per µl of blood at day 9 post treatment. $n = 6$ (nontreated, mock sgRNA(or $n = 7$ ($A_{2A}R$ sgRNA). **C** Gating strategy for intratumoral $CD8^+NGFR^+$ CAR T cells. **D, E** The expression of indicated markers on $CD8^+NGFR^+$ CAR T cells. Representative FACS plots shown from individual mice. For combined data, $n = 8$ (nontargeting sgRNA) or 7 ($A_{2A}R$ sgRNA). **F** The number of $CD8^+NGFR^+$ CAR T cells recovered from spleens or blood of treated mice at day 7 post treatment. $n = 7$ (nontargeting sgRNA, 8 $A_{2A}R$ sgRNA, or 5 shRNA $A_{2A}R$). **G** Comparison of the phenotype of $A_{2A}R$ knockdown and $A_{2A}R$ CRISPR/Cas9-edited CAR T cells at day 9 post treatment. **G** Proportion of $CD8^+NGFR^+$ CAR T cells that were CD62L-negative. $n = 7$ (nontargeting sgRNA, 8 $A_{2A}R$ sgRNA, or 5 shRNA $A_{2A}R$). For pooled data, values indicate the mean ± SEM. ***$p < 0.001$, **$p < 0.01$, *$p < 0.05$, one-way ANOVA. Source data are provided as a Source Data file.

characterized the effect of shRNA in modulating $A_{2A}R$-mediated suppression of CAR T cells[16,44], but these studies were limited to in vitro assessment of cytokine production and killing capacity. Importantly, no study has previously addressed the use of CRISPR/Cas9-mediated editing of $A_{2A}R$, which is likely to be the favored approach of gene disruption moving into the clinic. Therefore, to investigate this further, we developed approaches to target $A_{2A}R$ using either shRNA or CRISPR/Cas9 technology.

Our initial studies using CAR T cells derived from $A_{2A}R^{+/+}$, $A_{2A}R^{+/-}$, or $A_{2A}R^{-/-}$ mice indicated that $A_{2A}R^{+/-}$ CAR T cells retained high sensitivity to the adenosine analog NECA even at low doses. This suggests that T-cell effector function is highly sensitive to low-level $A_{2A}R$ expression and infers that either gene-based or pharmacological approaches for targeting $A_{2A}R$ should aim to achieve greater than 50% inhibition to mediate a significant reduction in adenosine sensitivity. Thus, despite high levels of transduction with shRNAs targeting $A_{2A}R$, and significant suppression of $A_{2A}R$ signaling, shRNA-mediated knockdown was unable to abrogate NECA-mediated suppression of CAR T-cell function. This was in contrast to CRISPR/Cas9-mediated deletion of $A_{2A}R$, which was highly efficient and led to a complete loss of $A_{2A}R$ signaling as shown by lack of cAMP accumulation and genome-wide transcriptional analyses. Our results are consistent with previous studies indicating that $A_{2A}R$ is the major target for adenosine on T cells and that other adenosine receptors such as the $A_{2B}R$ do not compensate for the loss of $A_{2A}R$[52]. This may be due, in part, to the observation that $A_{2A}R$ is required for expression of $A_{2B}R$ on the cell surface[53]. Nevertheless, these concepts support our observation that targeting the $A_{2A}R$ with CRISPR/Cas9 is sufficient to attenuate adenosine-mediated suppression of CAR T cells.

Curiously, shRNA-mediated knockdown of $A_{2A}R$ promoted effector T-cell differentiation, particularly in response to stimulation through the CAR. This led to increased effector function at the tumor site, including enhanced cytokine production and expression of both granzyme B and Ki-67. However, it is now appreciated that increased CAR T-cell activation can lead to enhanced terminal effector-like differentiation that can compromise persistence. Indeed, consistent with this, we observed a significant reduction in the number of CAR T cells recovered ex vivo following shRNA-mediated $A_{2A}R$ knockdown. By contrast, however, CRISPR/Cas9-mediated deletion of $A_{2A}R$ had minimal effect on CAR T-cell memory phenotype in both murine and human CAR T cells, leading to comparable persistence relative to control CAR T cells. This result is consistent with the observation that $A_{2A}R$ deletion promotes the persistence of $CD8^+$ T cells in the setting of LCMV[38] and our own previous observations using CAR T cells derived from $A_{2A}R$-deficient mice, which elicited comparable engraftment to their wild-type counterparts[16]. The reasons for these differences are not fully understood; however, a previous study has indicated a role for $A_{2A}R$ signaling in promoting T-cell memory, enhancing the expression of IL-7R[11], a key pro-survival factor. Our data suggest

that $A_{2A}R$ knockdown enhances CAR T-cell effector function at the expense of long-term CAR T-cell responses, whereas full knockout of $A_{2A}R$ circumvents the effects on T-cell persistence through unknown mechanisms. The reasons for this will be interesting to determine in future studies incorporating analysis of memory-associated genes in knockdown and knockout CAR T cells. One possibility that cannot be fully discounted however, is that the shRNAs elicited off-target effects that affected the memory response of CAR T cells in a nonspecific manner. However, as these effects were verified using independent shRNAs with distinct sequences, this would seem unlikely. Notably, comparisons between shRNA and CRISPR/Cas9 have shown that a significant proportion of shRNAs can elicit a distinct gene signature to CRISPR/Cas9 editing even when both technologies have been verified to be on-target[54].

Our data clearly show that CRISPR/Cas9 editing of $A_{2A}R$ is a very effective and potent method to enhance CAR T-cell function and render them resistant to adenosine, a major immunosuppressive factor. We first showed this using anti-human Her2-directed murine CAR T cells adoptively transferred into human Her2 transgenic mice. This model expresses human Her2 in the brain (cerebellum) and mammary tissue of mice allowing mice to tolerate the foreign human Her2 antigen[55]. Therefore, these studies highlight the capacity of $A_{2A}R$-edited CAR T cells to elicit improved antitumor immune responses in an immunocompetent model, which contains immunosuppressive subsets such as MDSCs and Tregs that produce adenosine and limit CAR T-cell function in solid tumors[56]. We also demonstrated that this approach was effective with human CAR T cells using the clinically relevant anti-Lewis Y CAR T-cell construct[40] (NCT03851146) whereby $A_{2A}R$-edited CAR T cells elicited significantly greater therapeutic effects. Deletion of $A_{2A}R$ by CRISPR/Cas9 editing significantly enhanced the transcription of a number of pro-inflammatory cytokines by CAR T cells. Among these, we validated that the production of IFNγ, TNF, and MIP1α was enhanced in $A_{2A}R$-edited CAR T cells at the protein level. Interestingly, while $A_{2A}R$ deletion completely attenuated the suppressive effect of NECA on IFNγ and MIP1α production, TNF remained partially suppressed by NECA in $A_{2A}R$-edited cells, albeit to a lesser extent. This could suggest a role for $A_{2B}R$ in adenosine-mediated inhibition of TNF production or residual $A_{2A}R$ expression on a subset of CAR T cells. The latter is consistent with the increased sensitivity of TNF to $A_{2A}R$-mediated expression as observed in heterozygous $A_{2A}R^{+/-}$ CAR T cells. Notably, we and others have previously identified that both IFNγ and TNF are critical for CAR T-cell effector function in vivo[16,32,57], underlining the importance of increasing the production of these cytokines. This is likely related to the capacity of IFNγ and TNF to evoke "bystander" activity on tumor cells not directly targeted by the CAR T cells[33], modulation of other immune cells within the tumor microenvironment, and/or its propensity to modulate the tumor stroma and inhibit neovascularization as previously observed in the context of $A_{2A}R$-deficient

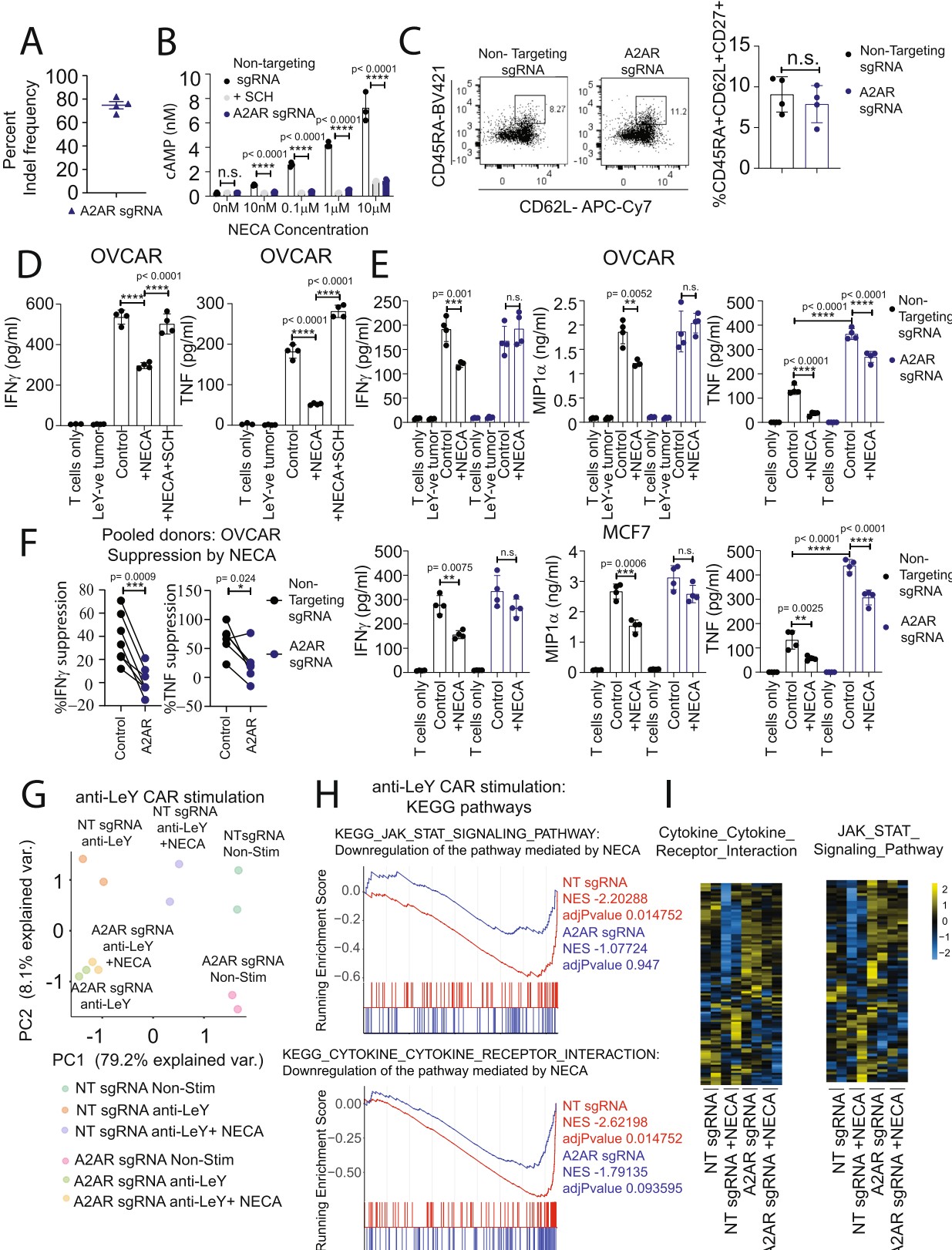

T cells[7]. Interestingly, we observed that $A_{2A}R$ deletion similarly enhanced the cytokine production of both $CD8^+$ and $CD4^+$ CAR T cells in vivo. This is of significance given emerging data showing the importance of $CD4^+$ T cells in the antitumor efficacy mediated by CAR T-cell therapy. In our studies, we observed that the deletion of $A_{2A}R$ in $CD8^+$ CAR T cells was sufficient to mediate an enhanced therapeutic response. However, these results do not preclude a role for $A_{2A}R$ deletion in $CD4^+$ CAR T cells, which may become more apparent when using a higher proportion of $CD4^+$ CAR T cells than used in our syngeneic model. Furthermore, while adenosine has been shown to suppress the effector function of both $CD8^+$ and $CD4^+$ conventional

**Fig. 7 Targeting $A_{2A}R$ by CRISPR/Cas9 enhances the effector function of human anti-Lewis Y CAR T cells.** Human anti-Lewis Y CAR T cells were generated and then treated with CRISPR/Cas9 and a sgRNA targeting either human $A_{2A}R$ or a nontargeting control. CAR T cells were then cultured for 6–9 days prior to downstream assays. **A** INDEL frequency as determined by ICE analysis. Data represent the mean ± SEM of four different donors. **B** cAMP accumulation following stimulation with indicated doses of NECA. Where indicated, SCH58261 (SCH, 1 μM) was added as a control. Data represent the mean ± SD of triplicate samples. Data from a representative experiment of $n = 2$. **C** Proportion of CD8$^+$ CAR T cells expressing a $T_{SCM}$ CD45RA$^+$CD62L$^+$CD27$^+$ phenotype. Left—representative experiment, Right—data represented as the mean ± SEM of four individual donors. Statistics determined using a paired $t$ test. **D**, **E**. $2 \times 10^5$ anti-Lewis Y CAR T cells were cocultured with MDA-MB-435 (Lewis Y negative, LeY$^-$), OVCAR-3, or MCF7 cells in the presence or absence of NECA (10 μM) or SCH58261 (10 μM). After 8 h, supernatants were harvested and production of IFNγ, TNF, or MIP1α determined. Data shown are the mean ± SD of triplicate/quadruplicate cultures from $n = 6$ experiments, ****$p < 0.0001$, ***$p < 0.001$, **$p < 0.01$, two-way ANOVA. **F** Percentage suppression of IFNγ and TNF production mediated by NECA following anti-Lewis Y CAR T-cell coculture with OVCAR-3 tumor cells. ***$p < 0.001$, *$p < 0.05$ paired $t$ test, $n = 6$ (TNF) or 7 (IFNγ) independent donors. **G**, **H** Anti-Lewis Y CAR T cells were stimulated with an anti-idiotype antibody (anti-LeY CAR) for 8 h in the presence or absence of 10 μM NECA and RNA analyzed by 3′RNA-Seq. **G** Principal component analysis based on top 100 most variable genes. **H** Gene set enrichment analyses for indicated pathways comparing anti-CAR-activated CAR T cells in the presence or absence of NECA (10 μM) for both nontargeting sgRNA and $A_{2A}R$ sgRNA-edited CAR T cells. **G**, **H** Statistical tests were performed with indicated R packages as outlined in the "Methods". **I** Heatmap of gene expression for pathways shown in (**H**). NT sgRNA—nontargeting sgRNA. Source data are provided as a Source Data file.

T cells[13], a recent study indicated that $A_{2A}R$ activation by inosine could enhance $T_H1$ differentiation[58]. Whether inosine can modulate the differentiation of CAR T cells remains unknown but could represent an interesting topic for further studies.

While we observed that $A_{2A}R$ activation suppressed cytokine production by both CD8$^+$ and CD4$^+$ CAR T cells, we did not observe a role for $A_{2A}R$ in suppressing the direct cytotoxic activity of anti-Lewis Y CAR T cells in these short-term assays. However, we cannot discount the possibility that long-term $A_{2A}R$ activation leads to impaired cytotoxic function in vivo, particularly given that $A_{2A}R$-edited CAR T cells displayed higher expression of granzyme B than control CAR T cells when isolated from tumors ex vivo.

Our data therefore support the concept of utilizing CRISPR/Cas9-mediated editing of $A_{2A}R$ to enhance CAR T-cell function. Our data support that a CRISPR/Cas9 strategy is likely more effective than either shRNA-mediated targeting or pharmacological blockade, likely related to the high affinity of the $A_{2A}R$ for adenosine, meaning that any residual $A_{2A}R$ receptor expression remains able to exert potent immunosuppression. However, it may be of interest to combine this approach with other strategies designed to target tissue hypoxia, such as supplemental oxygenation or hypoxia-sensitive prodrugs[59,60]. From a safety perspective, although there may be potential concerns with permanent $A_{2A}R$ deletion and excessive immune responses directed against the host, we observed that $A_{2A}R$-edited CAR T cells were safe and well tolerated as shown by unaltered homeostatic concentrations of enzymes that are indicative of liver or kidney toxicities and histology of tissue sections of mice that had undergone therapy. Indeed, targeting the $A_{2A}R$ is likely to have a favorable safety profile relative to other immunosuppressive pathways, given that the hypoxia–adenosine axis is more prominent in the tumor microenvironment. The clinical translation of this approach is therefore high, and notably, transient delivery of sgRNA/Cas9 RNP as used in this study has been shown to be highly effective and results in limited off-target editing[20]. Thus, this approach has high translational potential given the emergence of clinical trials supporting the use of ACT and CAR T cells in combination with immunotherapies designed to overcome tumor-induced immunosuppression (NCT03287817, NCT03296137, NCT03310619, and NCT02070406). Furthermore, CRISPR/Cas9-mediated editing of CAR T cells has precedence since CRISPR/Cas9-mediated deletion of various inhibitory genes, including PD-1, TGFβR, and PTPN2, has previously shown to be effective and PD-1-edited T cells have now been used in patients[17,19,20,39]. Given that $A_{2A}R$-edited CAR T cells will remain subject to these other immunosuppressive pathways, we believe that in future studies, it will be of interest to investigate targeting multiple suppressive genes using CRISPR/Cas9 to further protect CAR T cells against tumor-induced immunosuppression.

## Methods

**Cell lines and mouse models.** Murine tumor cell lines on the C57BL/6 background were used. E0771 (breast cancer), a gift from Prof. Robin Anderson (Olivia Newton John Cancer Research Institute), 24JK (sarcoma, Patrick Hwu, NIH, Bethesda, Maryland, USA), and MC38 (colon carcinoma, Dr. Jeff Schlom, NIH, Bethesda, Maryland, USA) were engineered to express truncated human HER2[61]. OVCAR-3, MCF7, and MDA-MB-435 were obtained from the American Type Culture Collection. PCR analysis was used to verify that tumor lines were mycoplasma-negative. Tumor cells were grown in RPMI supplemented with 10% FCS, 0.1 mM nonessential amino acids (NEAA), HEPES, 2 mM glutamax, 1 mM sodium pyruvate, and penicillin/streptomycin. For in vivo experiments, E0771-Her2 tumor cells were resuspended in PBS (20 μl) and injected into the fourth mammary fat pad. C57BL/6 WT mice or C57BL/6 hHer2 mice were bred at the Peter MacCallum Cancer Centre (PMCC) and used between 6 and 16 weeks of age. Animal experiments were approved by the Peter MacCallum Cancer Centre Animal Experimentation Ethics Committee, Project number #E582.

**Antibodies and cytokines.** For murine T-cell stimulation, anti-CD3 (clone 145-2C11) and anti-CD28 (clone 37.51) antibodies were purchased from BD Pharmingen. Human anti-CD3 (OKT3) was purchased from Ortho Biotech. Recombinant human IL-2/human IL-15 and murine IL-7 used for T-cell stimulation were obtained from the NIH and PeproTech, respectively. The antibody for stimulation of the anti-Her2 CAR (9B11) was obtained from Cell Signaling and used at a concentration of 1:5000. The purified anti-Idiotype Ab for stimulation of the anti-Lewis Y CAR (clone LMH3) was obtained from Andrew Scott (Olivia Newton John Cancer Research Institute). For cell stimulation, the antibody was plate-bound at a concentration of 900 ng/ml. Details on the antibodies used for flow cytometry are included in Supplementary Table 1.

**Generation of murine CAR T cells.** Retrovirus encoding the anti-Her2 CAR (CD28 and CD3ζ signaling domains) was obtained from supernatant derived from the GP + E86 packaging line as previously described[26,62]. This GP + E86 anti-Her2 CAR packaging line was also engineered to produce retrovirus encoding a small short-hairpin RNA either targeting $A_{2A}R$ or a scrambled, nontargeting control and a truncated human NGFR. Sequences for these shRNAs are as follows: Scramble shRNA TCTCGCTTGGGCGAGAGTAAGTAGTGAAGCCACAGATGTACTTA CTCTCGCCCAAGCGAGA, $A_{2A}R$ shRNA1 CTCATTAAGAGAAGAGAAGAAT AGTGAAGCCACAGATGTATTCTTCTCTTCTCTTAATGAG, $A_{2A}R$ shRNA 2 CTGCTCCAGAAGCGTCCACAATAGTGAAGCCACAGATGTATTGTGGAC GCTTCTGGAGCAG, $A_{2A}R$ shRNA 3 CTGGAAGACAGAAACTATGAATAGTG AAGCCACAGATGTATTCATAGTTTCTGTCTTCCAG.

Splenocytes were cultured in RPMI supplemented with 10% FCS, sodium pyruvate, glutamax, NEAA HEPES, and penicillin/streptomycin. Splenocytes were activated in media containing anti-CD3 (0.5 μg/ml, 1:1000), anti-CD28 (0.5 μg/ml, 1:1000) IL-2 (100 IU/ml), and IL-7 (200 pg/ml) at a concentration of $5 \times 10^6$ per milliliter. Twenty-four hours later, T cells were isolated following a Ficoll centrifugation. Retroviral supernatant was added to retronectin-coated (10 μg/ml) plates (Takara Bio, 4 ml per well of a 6-well plate). The supernatant was spun onto plates (1200g for 30 min) after which T cells, resuspended in IL-2/IL-7 containing retroviral supernatants, were added to the plates to give a final volume of 5 ml/well. The final T-cell concentration was between $7 \times 10^6$ and $10 \times 10^6$ per well. T cells were spun for 90 minutes (1200 g) and then incubated overnight before repeating

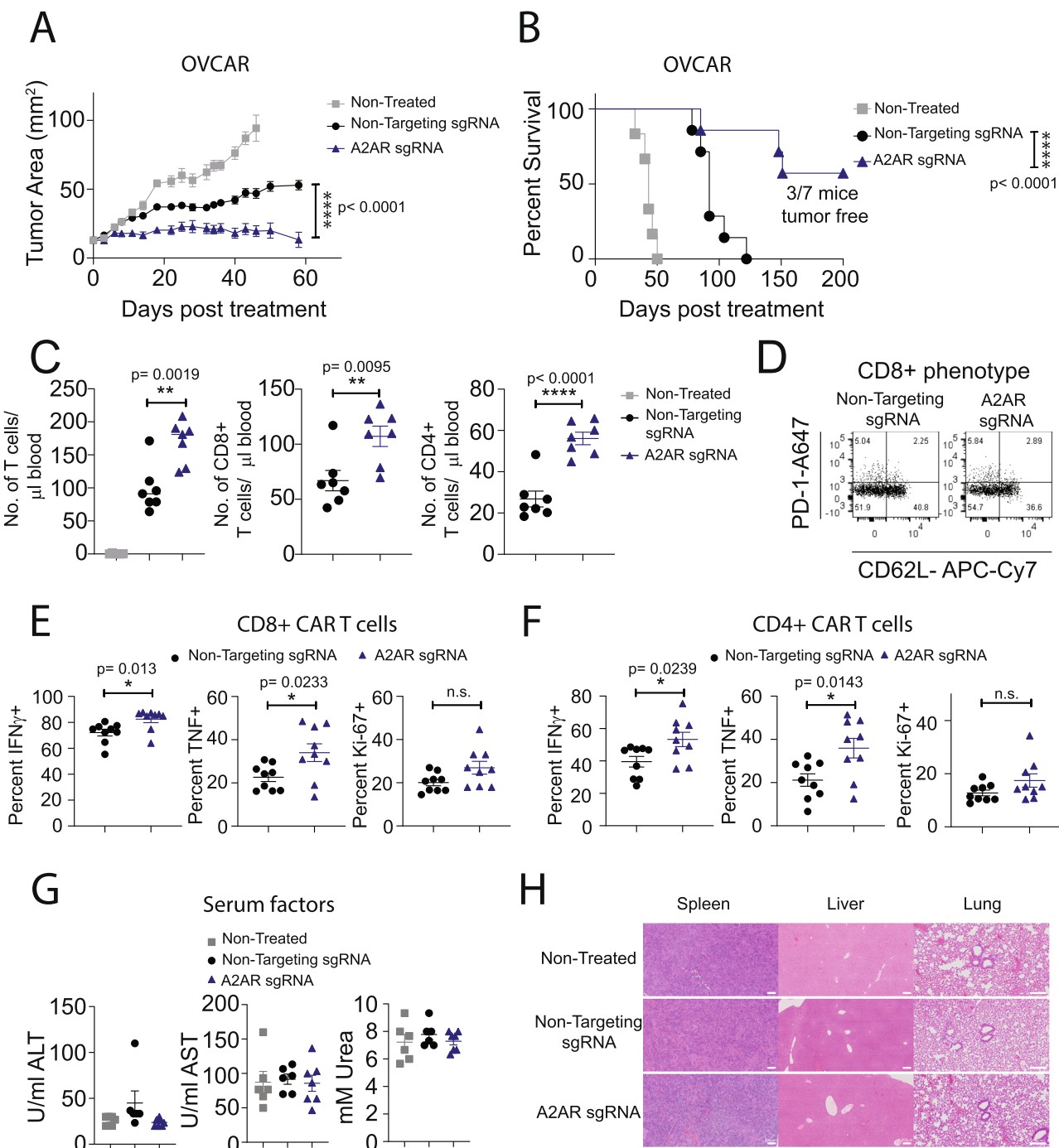

**Fig. 8 Targeting A$_{2A}$R by CRISPR/Cas9 enhances the in vivo efficacy of human anti-Lewis Y CAR T cells.** NSG mice were injected subcutaneously with 5 × 10$^6$ OVCAR-3 tumor cells. Once tumors were established (15–20 mm$^2$), mice were irradiated (1 Gy) and treated with 9 × 10$^6$ anti-Lewis Y CAR T cells generated as per Fig. 6 on two subsequent days. Mice were then treated with 50,000U of IL-2 on days 0–4 post CAR T cell treatment. **A** Tumor growth, data shown are the mean ± SEM of 6–7 mice per group, ****$p$ < 0.0001 per group, two-way ANOVA. **B** Survival determined as when tumors exceeded 80 mm$^2$. ****$p$ < 0.0001 Log-Rank test. **C, D** Mice were bled at day 8 post treatment. **C** The number of human T cells per μl of blood was determined. Data shown are the mean ± SEM, **$p$ < 0.01, ****$p$ < 0.0001 paired $t$ test. **A–C** $n$ = 6 (nontreated) or 7 (mock sgRNA or A$_{2A}$R sgRNA) mice per group. **D** The phenotype of CD8$^+$ CAR T cells in terms of CD62L and PD-1 expression. Data shown are concatenated from $n$ = 7 per group. **E, F**. At day 14 post treatment, tumors were excised and the proportion of (**E**) CD8$^+$ and (**F**) CD4$^+$ CAR T cells expressing IFNγ, TNF, and Ki-67 was determined. Data shown are the mean ± SEM of 9 mice per group. *$p$ < 0.05 paired $t$ test. **G** Serum was collected at day 40 post treatment and the concentration of defined serum factors determined. Data shown as the mean ± SEM of $n$ = 6 (nontreated, mock sgRNA) or 7 (A$_{2A}$R sgRNA) mice per group. **H** At the experimental endpoint, lungs, liver, and spleens were collected and sections analyzed by hematoxylin and eosin staining. One representative mouse of three per group is shown. A scale bar indicating a size of 200 μm is shown. Source data are provided as a Source Data file.

the transduction process. At the end of the transduction process, T cells were cultured in IL-2 and IL-7 or IL-7 and IL-15 (10 ng/ml) containing media. Transduced CAR T cells were used between days 6 and 8 after transduction.

**Human CAR T cells**. Ethics approval was granted by both the Red Cross and the Peter MacCallum Cancer Centre Human Research and Ethics committee (HREC# 01/14). Informed consent was obtained from blood donors by the Red Cross. Human peripheral blood mononuclear cells (PBMCs) were isolated from normal buffy coats using Ficoll centrifugation and PBMCs were cultured in RPMI supplemented with 10% FCS, sodium pyruvate, glutamax, NEAA, HEPES, and penicillin/streptomycin. PBMCs were activated with anti-human CD3 (30 ng/ ml, 1:33,000) and IL-2 (600 IU/ml) for 48 h before transduction with a retrovirus encoding a second-generation scFv–anti-Lewis-Y CAR (CD28 and CD3ζ). Retroviral supernatant was obtained from a modified PG13 packaging line that produces a CAR with an scFv generated from the humanized monoclonal antibody Hu3S193[41] and 5 ml added to retronectin-coated (15 μg/ml) 6-well plates that were centrifuged at 1200 g for 1 hr. Activated PBMCs were resuspended at a concentration of $0.5 \times 10^6$ per ml in fresh 5 ml of retrovirus containing IL-2 and added to the retronectin-coated plates following the removal of the viral supernatant used during the coating phase. Cells were centrifuged for 1 h (1200 g) followed by overnight incubation and a second round of transduction the next day. Cells were cultured at a density of $1.5 \times 10^6$/ml following the transduction process.

**CRISPR/Cas9 gene editing**. Human CAR T cells were edited after activation and transduction with CAR as described above, whereas murine T cells were edited as freshly isolated splenocytes prior to activation. Electroporation of T cells was performed as previously described[63]. Briefly, per $1 \times 10^6$ activated human T cells or $20 \times 10^6$ murine splenocytes to be electroporated, 37 pmoles recombinant Cas9 (IDT) and 270 pmoles sgRNA (Synthego) were combined and incubated for 10 min at RT to generate Cas9/sgRNA RNP. Human T cells ($1 \times 10^6$) or murine T cells ($20 \times 10^6$) were resuspended in 20 μL P3 Buffer (Lonza), combined with RNP and electroporated using a 4D-Nucleofector (Lonza) using pulse code E0115 and DN100 for human and mouse T cells, respectively. Prewarmed complete RPMI media was then immediately added to electroporation wells and T cells were allowed to recover for 10 min at 37 °C. Murine T cells were subsequently activated overnight and subjected to CAR transduction (as above) and human T cells were expanded using a previously defined rapid expansion protocol[16]. sgRNA sequences used were *Thy1* GGAGAGCGACGCUGAUGGCU *Adora2a*: UGUCGAUGGCA AUAGCCAAG, *ADORA2A*: CUACUUUGUGGUGUCACUGG, nontargeting control: GCACUACCAGAGCUAACUCA. Estimation of INDEL frequency in the *Adora2a* gene was performed by PCR amplifying the sgRNA target site with primers 5'-TGCAGAACGTCACCAACTTC-3' and 5'-CAGCCTCGACATGTGAC CTA-3' in control or CRISPR-edited murine CAR T cells. A similar analysis was performed in human CAR T cells with primers 5'-ATCATGGGCTCCTCGGTG TAC-3' and 5'-CTGAAGATGGAGCTCTGCGTG-3'. The regions were Sanger sequenced and INDEL frequency was estimated using online ICE analysis (https://ice.synthego.com). Prediction of potential off-target cutting sites of the ADORA2A targeting sgRNA was performed using the COSMID tool[64]. Predicted off-target regions were PC-amplified from gDNA of edited human T cells (or unedited controls) using primers designed by COSMID, outlined in Supplementary Fig. 7A. Amplified fragments were Sanger sequenced and indel frequency was calculated using the ICE tool.

**cAMP assays**. A serial half-log dilution of 4x NECA (50-(N-Ethylcarboxamido) adenosine, Abcam) was prepared in stimulation buffer (HBSS containing 5 mM HEPES, 0.1% bovine serum albumin, 25 μM rolipram, pH adjusted to 7.4). Similarly, a standard curve was generated with a serial half-log dilution of provided cAMP standard from the LANCE ultra cAMP kit (PerkinElmer, catalog no. TRF0262). In all, 5 μL and 10 μL of 2× NECA and 1× cAMP standard, respectively, were added to wells prior to addition of cells. CAR T cells were harvested from culture, washed twice, and resuspended to $1 \times 10^6$ cells/mL in stimulation buffer. CAR T cells were preincubated with 1 μM of Forskolin (Abcam), with or without 1 μM SCH58261 (Abcam) for a total of 30 min before transfer of 5 μL of cell suspension to each well of a 384-well plate containing 2x NECA, thus adjusting the final volume to 10 μL/well. Cells were then incubated with drugs for another 30 min at room temperature. 4× Eu-cAMP tracer and 4× U-light anti-cAMP antibody were prepared in provided detection buffer at 1:50 and 1:150 dilution, respectively, and 5 μL of Eu-cAMP mix and 5 μL U-light mix were then added in that order to each well of standard curve and cell suspension. Plates were then incubated for a minimum of 1 h at room temperature and protected from light before storing at 4 °C prior to measurement on an EnVision Multimode plate reader system (Perkin Elmer) using the Wallac Envision Manager v1.12 software. The TR-FRET signal (665 nm) was plotted against the concentration of cAMP to generate a nonlinear standard curve. Experimental cAMP concentrations were then interpolated from the standard curve.

**Nanostring analysis**. RNA prepared from stimulated CAR T cells was quantified and quality assessed via Tapestation (Agilent). Gene count were determined by the Peter MacCallum Cancer Centre Advanced Genomics Core using an nCounter XT assay (Mouse Pan Cancer Immune Profiling; NanoString Technologies 115000142) and analyzed by nSolver 4.0 software as per the manufacturer's instructions. Default quality control and normalization steps showed all samples were adequate for further analysis. Heatmaps were generated using the pheatmap R package, using row mean-centered and scaled gene expression of normalized log2 counts derived from nSolver. Rows were grouped by hierarchical clustering using Euclidean distance and average-linkage.

**Treatment of tumor-bearing mice**. Female C57BL/6 human Her2 transgenic mice were injected under anesthesia in the fourth mammary fat pad with $2 \times 10^5$ E0771-Her2 cells. Once tumors were established (day 6–7), mice were preconditioned with 4-Gy total-body irradiation (TBI) prior to the intravenous (i.v.) administration of $1 \times 10^7$ CAR T cells on two subsequent days. Mice were also treated intraperitoneally (i.p.) with 50,000 IU IL-2 on days 0–4 after T-cell transfer. For experiments with human anti-Lewis Y CAR T cells, NOD.Cg-Prkdcscid Il2rgtm1Wjl/SzJ (NOD scid gamma, NSG) mice were injected subcutaneously with $5 \times 10^6$ OVCAR-3 tumor cells, and once established (~20 mm$^2$) mice were preconditioned with 1-Gy TBI and i.v. injected with $6–9 \times 10^6$ CAR T cells on two consecutive days. Following T-cell transfer, mice were treated with 50,000 IU IL-2 I.P. on days 0–4.

**Analysis of tumor-infiltrating immune subsets**. To analyze CAR T cells infiltrating tumors by flow cytometry, mice were euthanized and tumors excised and digested using DMEM supplemented with 1 mg/ml collagenase type IV (Sigma-Aldrich) and 0.02 mg/ml DNAase (Sigma-Aldrich). Tumors were incubated in this medium for 30 min at 37 °C, rendered into single-cell suspensions and then filtered twice (70 μm). Cells were then analyzed by flow cytometry directly ex vivo. For analysis of cytokines, single-cell suspensions were stimulated with PMA (5 ng/ml) and ionomycin (1 μg/ml, Sigma-Aldrich) with GolgiPlug and GolgiStop (BD Biosciences) for 3 h prior to downstream flow cytometry staining and analysis. Flow cytometry analysis was performed on FlowJo v10.5.0.

**RNA-sequencing analysis**. RNA-seq libraries were prepared from RNA using the Quant-seq 3' mRNA-seq Library Prep Kit for Illumina (Lexogen) as per the manufacturer's instructions. NextSeq (Illumina, Inc., San Diego, CA) sequencing was performed to generate single-end, 75-bp RNA-seq short reads and subsequently CASAVA 1.8.2 was used for base calling. The quality of output was assessed using RNA-SeQC v1.1.7[65]. The data were then quality-trimmed (Cutadapt v1.6) to remove random primer bias and 3' end trimming was performed to remove poly-A-tail derived reads. Sequence alignment against the mouse reference genome mm10 or the human genome hg38 was performed using HISAT2. Finally, the subread software package 1.6.4 was used to count the number of raw reads per gene using gene definitions from Ensembl Release 96[66]. Gene counts were normalized using the TMM (trimmed means of $M$-values) method and converted into log2 counts per million (CPM) using the EdgeR v 3.8.5 package[67,68]. For differential gene expression between groups, the quasi-likelihood F test statistical test method based on the generalized linear model (glm) framework from EdgeR was used. Principal component analysis (PCA) was performed on normalized counts and EdgeR or Voom based on most variable genes. Adjusted p values were computed using the Benjamini–Hochberg method. All differentially expressed genes were classified as significant based on a false discovery rate (FDR) cutoff of 5%. Volcano plots were used to represent differential gene expression between groups.. Using row mean-centered and scaled normalized log2(CPM + 0.5), heatmaps were generated using the pheatmap R package. Rows were grouped by unbiased hierarchical clustering using Euclidean distance and average-linkage. Each group was sequenced in biological duplicates or triplicates and mean of CPM values was used.

The murine adenosine signature was generated based on genes that were significantly (FDR < 0.05) up- or downregulated in three independent experiments following NECA treatment of anti-myc Tag stimulated murine anti-Her2 CAR T cells. Similarly, for the human adenosine signature, a gene signature was created based on a threshold of FDR < 0.05 and FC greater than 1 for significantly upregulated, or less than 1 for significantly downregulated genes were identified from anti-LeY stimulated CAR T cells treated with NECA compared to control.

Unbiased gene set enrichment analysis was performed using fgsea package (v3.12) on differential expressed genes preranked by fold change with 1000 permutations (nominal *P*-value cutoff < 0.05)[69]. Reference gene sets were obtained from the MsigDB library, canonical pathways: KEGG (https://www.genome.jp/kegg/kegg1.html)[70–73].

**Assessment of toxicity in treated mice**. Sera were assessed by the Peter MacCallum Cancer Centre Pathology department for parameters, including ALT, AST, and urea. For analysis of serum cytokines, sera were diluted 1:3 and analyzed by cytometric bead array kit (BD) according to the manufacturer's protocol. Beads were acquired using FACSVerse and analyzed using FCAP Array (version 3) software (BD). For histology, organs were incubated in neutral buffered formalin for at least 24 hours before replacement with 70% ethanol prior to processing by the Peter MacCallum Cancer Centre's Centre for Imaging and Advanced Microscopy. Organs were processed, paraffin-embedded, and sectioned to 3-μm thickness prior to hematoxylin and eosin staining. Images were acquired using an

Olympus VS120 microscope under 40× magnification and images analyzed using ImageJ software (v 1.52u).

**Statistics**. Statistical tests were performed as described in the figure legends with a $P$ value of less than 0.05 considered significant. All $t$ tests and log-rank tests were two-tailed. The program Graphpad Prism v8.0.2 was used.

**Reporting summary**. Further information on research design is available in the Nature Research Reporting Summary linked to this article.

## Data availability

The RNA-sequencing data that support the findings of this study have been deposited in GEO NCBI under the accession code GSE156192 that contains the subseries GSE156189, GSE156190, GSE156191, and GSE166807. KEGG datasets utilized can be accessed via https://www.gsea-msigdb.org/gsea/msigdb/index.jsp. Source data are provided with this paper. The remaining data are available within the paper, supplementary information, and tables or available upon request from the authors. Source data are provided with this paper.

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

## Acknowledgements

This work was funded by a Program Grant from the National Health and Medical Research Council (NHMRC, Grant number 1132373), an NHMRC Project grant (APP1122444), and a Synthego Genome Engineer Innovation Grant. J. Lai is supported by Cancer Research Institute Irvington Postdoctoral Fellowship (CRI Award #3530). P. A. Beavis was supported by a National Breast Cancer Foundation Fellowship (IECF-17-005; 2017–2020) and a Victorian Cancer Agency Mid-Career Fellowship (MCRF20011, 2021–current). I.G. House is supported by a Victorian Cancer Agency Early Career Fellowship (ECRF20017). P.K. Darcy is supported by an NHMRC Senior Research Fellowship (APP1136680). The authors wish to acknowledge the contribution of consumer representatives Karen Gill, Mike Rear, and Graeme Sissing for their contribution to the study and research direction of the laboratory.

## Author contributions

L.G., K.S., I.G.H., P.K.D., and P.A.B. designed the experiments, developed the methodology, interpreted the data, and wrote the paper. A.X.Y.C., C.M., and G.S.D. also developed the methodology. L.G., K.S., M.A.H., J.L., A.X.Y.C., D.M., K.L.T., E.V.P., S.M., I.G.H., P.K.D., and P.A.B. performed the experiments and acquired the data. C.M., G.D.S., B.J.S, I.A.P., P.J.N., S.J.H., and L.M.K. provided the technical assistance and advice on the data analysis and interpretation. I.G.H., P.K.D., and P.A.B. supervised the study and were responsible for coordination and strategy.

## Competing interests

S.J.H. declares the following competing interests: Consultancy: Amgen, Celgene, Novartis, Janssen-Cilag, Haemalogix, GSK. Research funding: Celgene, Janssen-Cilag, and Novartis. L.M.K declares the following conflicts: consultancy and research funding from Agios and Celgene. P.K.D. declares the following conflicts: research funding from Myeloid Therapeutics, Prescient Therapeutics, and Juno Therapeutics. P.A.B. declares the following conflicts: research funding from AstraZeneca and Gilead Sciences. The remaining authors declare no competing interests.
