## [Peer Review File · Nature Communications]

REVIEWER COMMENTS

Reviewer #1 (Remarks to the Author): with expertise in CAR-T

Giuffrida et al report the CRISPR gene editing of CAR-T cells to delete the adenosine receptor with the final goal to improve the antitumor effects of these cells in the presence of adenosine. The topic and the technology proposed is not particularly novel, but it has some value for the clinical translation. However, many experiments lack in appropriate controls and the antitumor effects in immunocompetent mice is not particularly strong. Furthermore, the NSG model to test human CAR-T cells is not particularly appropriate to test the hypothesis and at the moment it remains unclear why gene edited CAR-T cells are performing better in NSG mice as compared to CAR-T cells.

Specific comments

- 1) In Figure 1, the authors must show if CAR-T cells in WT, Het, and KO T cells are eliminating tumor cells in vitro, and if NECA is affecting the tumor elimination in WT and Het T cells versus KO T cells in coculture experiments. The reduction of IFN γ and TNF α secretion by CAR-T cells is not directly indicating an impairment of their capacity to eliminate tumor cells via granzyme-B/perforin in in vitro coculture experiments. Similar comments apply for the in vitro experiments described in Figure 2 and 4.
- 2) The experiment in vivo in Figure 4 lacks the control of siRNA engineered CAR-T cells. Even if it is not correct to compare experiments in two different figures, the antitumor effect of shRNA2-A2CAR seems pretty similar to that caused by A2AR sgRNA. A side by side comparison in the same experiment is recommended. Similarly, CAR-T cells plus SCH58261 were compared in a separated experiment without CAR-T cells genetically modified to KO A2AR. Finally, the proposed genetic modification does not eradicate the tumor, but transiently delays tumor progression. This modest effect should be mechanistically explained or compared with other genetic modifications proposed to counterattack the tumor microenvironment.
- 3) As indicated in the comments for Fig.4, in Fig.5 there is no control of mice treated with siRNA CAR-T cells.
- 4) As mentioned in comment 1, also the experiments in vitro with human CAR-T cells do not show if NECA impairs the capacity of CAR-T cells to eliminate tumor cells in vitro in coculture experiments using CAR-T cells and tumor cells at different ratios. Short term assays such as a 4 hour Cr51 release assay are not really informative.
- 5) The antitumor effect of gene edited CAR-T cells in the NSG mouse model is more significant even if it is limited to one single tumor model. However, in this immunodeficient model it remains difficult to assess which is the role of adenosine since there is no tumor microenvironment in these tumor models. The observed antitumor effects may be simply determined by enhanced effector function of gene edited CAR-T cells rather than enhanced persistence at the tumor site which is the main goal of CAR-T cells in solid tumors. Re-challenge experiments should be performed to assess persistence of CAR-T cells in this model. Furthermore, a PDX model knowing to recruit murine cells to form the stroma would be helpful. In addition, the control with SCH58261 should be added in these experiments.

Reviewer #2 (Remarks to the Author): with expertise in CAR-T

This manuscript from Giuffrida and colleagues describes enhancement of CAR-T cell potency in mouse tumor models by CRISPR/Cas9 editing of the adenosine A2AR receptor. They show mechanistic studies indicating that the A2AR –edited cells were resistant to adenosine-mediated transcriptional changes, with increased cytokine production and expression of JAK-STAT signaling pathway genes. These studies are detailed and well done, and make a persuasive case for trials testing the editing of A2AR as a strategy to improve CAR-T function in humans.

Major point:

The authors extensively test the effect of knocking down or editing the A2AR gene on CAR-T function using both mouse and human CAR-T and in cell culture and in mouse models and using shRNA knockdown and CRISPR/Cas9 editing. The effect is too much data making the same point.

The ineffective knock down by shRNA is by now conventional wisdom in the field and their conclusion that that the shRNA mediated knock down is inferior to CRISPR/Cas9 knock out is unsurprising, This reviewer strongly feels that the shRNA experiments are dispensable to this paper and that it would benefit from editing for length and to focus on the CRISPR/Cas9 experiments. Further, a more limited version of the shRNA work has been reported twice, by these investigators and another group.

Minor point:

Was any effort made to evaluate off-target editing in any of these CAR-T cells? GUIDEseq or similar would be useful to confirm specificity of editing the ADORA2A gene.

Reviewer #3 (Remarks to the Author): with expertise in CAR-T and genome editing

Adenosine is an immunosuppressive factor that limits anti-tumor immunity through the suppression of multiple immune subsets including T cells via activation of the adenosine A2AR receptor (A2AR). In their previous work that was published in JCI in 2017, the authors showed that either genetic (A2AR-targeted shRNA) or pharmacological targeting of the A2AR profoundly increased CAR T cell efficacy. In this study, they claim that CRISPR/Cas9 mediated gene deletion of A2AR were superior to shRNA mediated knockdown or pharmacological blockade of A2AR. However, the study was separated into two parts, one for shRNA mediated knockdown of A2AR and another one for CRISPR/CAS9 mediated knockdown of A2AR. The conclusions were drawn from comparing two parts of the work. There was no any single experiment that compared shRNA mediated knockdown of A2AR and CRISPR/CAS9 mediated knockdown of A2AR directly, which is very unusual and the conclusions made from such study design is unacceptable. To make the study valuable with novelty, at least a control group of shRNA mediated knockdown CART has to be included in both Figure 4 and Figure 5.

Reviewer #4 (Remarks to the Author): with expertise in adenosine signaling

In this study Giuffrida...Beavis et al. investigated whether silencing/deletion of A2AR through CRISPR/Cas9-mediated approach is superior in reversing A2AR-mediated suppression of T cells for CAR-T therapy of solid tumors as compared with another commonly used approach, shRNA-mediated silencing. Although the data is encouraging, in its current form the study needs additional data and the text need additional updates to support the main conclusion and present a better view of the findings. Specifically;

1. CAR-T cells generated by CRISPR/Cas9-mediated deletion and shRNA-mediated silencing A2AR should be compared in the same tumor growth experiment along with immunophenotyping in both mouse and human settings.
2. The way CAR-Ts are prepared for the two approaches is different: A2AR is deleted by CRISPR/Cas9 before expansion and in another setting T cells are expanded before silencing. If possible, using similar approaches for both settings will better support the conclusion.
3. When human CAR-Ts are prepared, cells are activated before the deletion of A2AR meaning they may already get A2AR signaling along with IL-7/IL-2 signaling until editing with CRISPR/Cas9. In fact, in murine setting CAR-T cells show more effector/exhausted phenotype (more GRZB, PD1 and Tim3, Less CD62L-Figure 5) while that is not the case for human CAR-Ts (Figure 6-7)
4. One major effect of A2AR-mediated immunosuppression is downregulation of IL-2 receptor and IL-2, suggesting the help from CD4 T cells maintaining CD8 T cells in the absence of A2AR is important. Since the study is very similar to Beavis et al. (PMID: 28165340) one interesting and important addition would be testing the importance of deletion of A2AR in CD4s vs CD8 vs. both to provide a more mechanistic view other than classical in vitro cAMP-derived suppression mechanism which may not reflect what is happening in vivo.

5. A recent study by Mager et al. suggests that microbiome-derived inosine stimulates A2AR and promotes checkpoint responsiveness. One potential mechanism is A2AR-signaling promoting growth factor responsiveness of T cells. Since CAR-T cells constitutively express CARs they receive CAR-mediated continuous survival signals as opposed to endogenous/antigen-specific conventional T cells, which downregulate their TCRs upon activation and generally do not receive a strong TCR signal. In conventional T cell settings (even in CAR-T setting) all the activation markers presented here is a sign of terminal/effector T cell differentiation and lack of efficacy. This point should be discussed along with proper citation of relevant references (some examples are: PMID: 32792462; PMID: 32792462; PMID: 15980149)

Response to Reviewer Comments

We wish to thank the reviewers for their positive appraisal of our study and for their recommendations to further improve the quality of the manuscript. To address the reviewers' concerns we have added 26 new Figure panels (**Figures 4H, 4I, 5A-G, 6F, 6G, 7A, 8E, 8F Supplementary Figures 1A, 1D, 1E, 4D, 4E, 4G, 5E, 6A-C, 7A, 7D**) and modified the text as appropriate.

We have attached a point by point response to each of the reviewers' comments and hope that the revised manuscript is now suitable for publication in *Nature Communications*.

Reviewer 1:

1) In Figure 1, the authors must show if CAR-T cells in WT, Het, and KO T cells are eliminating tumor cells *in vitro*, and if NECA is affecting the tumor elimination in WT and Het T cells versus KO T cells in coculture experiments. The reduction of IFN γ and TNF α secretion by CAR-T cells is not directly indicating an impairment of their capacity to eliminate tumor cells via granzyme-B/perforin in *in vitro* coculture experiments.

Response: We thank the reviewer for this suggestion. In these experiments, we used IFN γ and TNF production as a measure of A_{2A}R activity as our previous data indicates that adenosine potently suppresses cytokine production of T cells but has only a mild suppressive effect on direct (perforin/granzyme-mediated) cytotoxic activity *in vitro* (PMID 25672397). Notably, cytokine production is an important facet of CAR T cell function *in vivo*, allowing CAR T cells to kill tumor cells indirectly and modifying the tumor microenvironment to promote enhanced anti-tumor immunity. To address this in the context of CAR T cells in the current manuscript we have performed additional short and long-term killing assays with both murine and human CRISPR/Cas9-edited CAR T cells. These experiments show that activation of the A_{2A}R does not significantly affect direct cytotoxic activity of CAR T cells *in vitro*. These data support our original conclusions that the immunosuppressive effects of A_{2A}R are predominantly mediated through suppression of CAR T cell mediated cytokine production.

The role of cytokines in the *in vivo* anti-tumor activity of CAR T cells is well known since we (PMID 30651288) and others (PMIDs 32566933, 32803171) have shown that cytokines can evoke killing activity against tumor cells not directly targeted by CAR T cells. We provide new data highlighting that A_{2A}R-mediated suppression of CAR T cell cytokine production significantly reduces this cytokine-mediated killing effect. All together, the additional data addressing this topic is shown in the manuscript in **New Supplementary Figures 1A, 1D, 1E, 4D, 4E and 5E**. We would like to note that we do not discount a role for A_{2A}R activation in suppressing perforin/granzyme B-mediated killing *in vivo* as discussed below in response to comment 6.

With regards to A_{2A}R-mediated suppression of cytokine-mediated killing, our data in **New Supplementary Figure 1E** shows that supernatants derived from NECA treated CAR T cells exhibit

significantly reduced cytokine-mediated cytotoxic activity against tumor cells. This new data is referred to on line 106:

“A major immunosuppressive phenotype mediated by A_{2A}R activation is impaired cytokine production by T cells whilst cytotoxic activity is less affected^{16, 29}. As expected, while NECA had no significant effect on direct CAR T cell cytotoxic activity (**Supplementary Figure 1A**), it significantly suppressed IFN γ and TNF production by wild-type A_{2A}R^{+/+} but not A_{2A}R^{-/-} CAR T cells following coculture with either E0771-Her2 (**Figure 1E-F**) or 24JK-Her2 tumor cells (**Supplementary Figure 1B-C**). The role of these cytokines in the indirect killing of tumor cells has previously been shown^{30, 31, 32} and notably the addition of NECA led to a significant reduction in the cytokine-mediated killing activity of supernatants derived from these cocultures (**Supplementary Figure 1D-E**).”

2) Similar comments apply for the in vitro experiments described in Figure 2 and 4.

Response: We have performed additional cytotoxicity assays with murine CAR T cells following CRISPR/Cas9-mediated editing of A_{2A}R. In agreement with our new data in response to comment 1, we observed that deletion of the A_{2A}R does not significantly enhance the cytotoxicity of CAR T cells. This new data is shown in **Supplementary Figure 4D-E** and is referred to in the following text on line 201:

“However, CRISPR/Cas9-mediated knockout of A_{2A}R did not enhance the cytotoxicity of CAR T cells measured at either 4 or 16 hours, consistent with our previous observations indicating that short-term A_{2A}R activation modulates CAR T cell cytokine production but not cytotoxic function *in vitro* (**Supplementary Figure 4D-E**)¹⁶.”

3) The experiment in vivo in Figure 4 lacks the control of siRNA engineered CAR-T cells. Even if it is not correct to compare experiments in two different figures, the antitumor effect of shRNA2-A2CAR seems pretty similar to that caused by A2AR sgRNA. A side by side comparison in the same experiment is recommended. Similarly, CAR-T cells plus SCH58261 were compared in a separated experiment without CAR-T cells genetically modified to KO A2AR.

Response: We agree with the reviewer that this is an important experiment and to address this we have performed additional experiments that indicate in head to head experiments that CAR T cells subjected to CRISPR/Cas9 mediated deletion of the A_{2A}R elicits enhanced therapeutic effects (**New Figure 4I**) and increased persistence in the tumor, spleen and blood relative to CAR T cells modulated with A_{2A}R targeting shRNA (**New Figure 6F**). The enhanced persistence of A_{2A}R deficient (CRISPR/Cas9 edited) relative to A_{2A}R knockdown (shRNA edited) CAR T cells was associated with enhanced expression of CD62L in the spleen in CRISPR/Cas9-edited CAR T cells (**New Figure 6G**). In **Figure 4I** the therapeutic effects of CRISPR/Cas9-edited cells were also directly compared to the combination of CAR T cells and SCH58261 as requested. This new data is referred to in the following text on lines 214 and 251:

“Notably these therapeutic effects were more pronounced than those observed with either shRNA mediated knockdown of A_{2A}R or with pharmacological-mediated A_{2A}R blockade (**Figure 4I, Supplementary Figure 4F**).”

“Notably, a head to head comparison revealed that the number of CAR T cells recovered from the spleens at day 9 post treatment was significantly greater following CRISPR/Cas9-mediated deletion of A_{2A}R when compared to A_{2A}R-mediated knockdown via shRNA (**Figure 6F**). Furthermore, shRNA mediated knockdown of A_{2A}R was associated with reduced expression of CD62L, a marker of central memory T

Peter MacCallum Cancer Centre
305 Grattan Street
Melbourne Victoria
3000 Australia

Phone +61 3 8559 5000
Fax +61 3 03 8559 7379
ABN 42 100 504 883
petermac.org

Postal Address
Locked Bag 1 A'Beckett Street
Victoria 8006 Australia

cells, potentially explaining the reduced persistence relative to CRISPR/Cas9 edited CAR T cells (**Figure 6G**).”

4) Finally, the proposed genetic modification does not eradicate the tumor, but transiently delays tumor progression. This modest effect should be mechanistically explained or compared with other genetic modifications proposed to counterattack the tumor microenvironment.

Response: The E0771-Her2 model used in these studies is aggressive and so we strongly disagree that this effect is modest. In fact these effects observed are similar to the enhanced therapeutic activity we have historically observed with PD-1 blockade in this model (PMIDs 23873688, 32735774).

To directly compare targeting of the A_{2A}R with PD-1 we examined the efficacy of CAR T cells following CRISPR/Cas9-mediated deletion of A_{2A}R or PD-1 in anti-Her2 CAR T cells. We validated that this guide could successfully induce a PD-1 knockout phenotype (data not shown) and then compared the efficacy of PD-1 and A_{2A}R-edited anti-Her2 CAR T cells against established E0771-Her2 tumors (**Reviewer Figure 1A**). We observed comparable anti-tumor activity and a similar survival benefit through these two approaches. Moreover, a mechanistic analysis of intratumoural CAR T cells revealed that A_{2A}R and PD-1 editing were similarly effective in inducing an enhanced activation phenotype (**Reviewer Figure 1B**). These data are included below as Reviewer Only Figures but can be included in the manuscript upon request.

We believe these data indicate that the lack of curative responses reflects the aggressive nature of the E0771-Her2 model and that the anti-tumor effects of A_{2A}R editing in CAR T cells are on par with PD-1 deletion and are clinically significant. We note that curative responses are observed in the OVCAR-3 model (**Figure 8**), utilizing human A_{2A}R-edited anti-Lewis Y CAR T cells.

A**B**
Reviewer Figure 1 – CRISPR/Cas9-mediated deletion of the A_{2A}R enhances CAR T cell effector function comparably to PD-1 deletion

C57BL/6 Her2 mice were injected with 2×10^5 E0771-Her2 tumor cells into the fourth mammary fat pad and treated with anti-Her2 CAR T cells following CRISPR/Cas9 mediated editing of T cells with a sgRNA targeting PD-1 or A_{2A}R (or a non-targeting control). CAR T cells were cultured in IL-7 and IL-15 and once tumors were established, 1×10^7 CAR T cells were injected intravenously on two subsequent days following 4 Gy total body irradiation. Mice were treated with 50,000U of IL-2 on days 0-4 post CAR T cell treatment. **A.** Tumor growth shown as the mean \pm SEM of $n = 5-6$ mice per group. **B.** At day 9 post treatment, the phenotype of tumor-infiltrating CD8⁺NGFR⁺ CAR T cells were analyzed by flow cytometry. Data represented as the mean \pm SEM of $n = 6-9$ per group.

Postal Address

Locked Bag 1 A'Beckett Street
Victoria 8006 Australia

5) As indicated in the comments for Fig.4, in Fig.5 there is no control of mice treated with siRNA CAR-T cells.

Response: We agree with the reviewer that this is an important experiment to perform and this has now been conducted. Please refer to our response to comment number 3.

6) As mentioned in comment 1, also the experiments in vitro with human CAR-T cells do not show if NECA impairs the capacity of CAR-T cells to eliminate tumor cells in vitro in coculture experiments using CAR-T cells and tumor cells at different ratios. Short term assays such as a 4 hour Cr51 release assay are not really informative.

Response: To address this comment we have performed longer term killing assays (16 hours). Consistent with our murine data these experiments confirm that the predominant effect of A_{2A}R activation is upon cytokine production rather than cytotoxic activity. This data is shown in **New Supplementary Figure 5E** and indicates that while NECA mediates a modest suppression of cytotoxic activity this does not reach statistical significance. This mild suppressive effect mediated by NECA was less pronounced when using A_{2A}R-edited CAR T cells as would be expected. This data is referred to on line 281 of the revised manuscript:

“With anti-Lewis Y CAR T cells generated across multiple donors, the deletion of A_{2A}R significantly reduced the suppressive effect of NECA on both IFN γ and TNF (**Figure 7F**). By contrast, neither NECA, nor A_{2A}R deletion significantly modulated the cytotoxic capacity of CAR T cells upon coculture with OVCAR-3 tumor cells in either 4 hour (short-term) or 16 hour (long-term) chromium release assays (**Supplementary Figure 5C-E**).”

In addition we also performed long-term killing assays with murine anti-Her2 CAR T cells cocultured with MC38-Her2 tumor cells. Consistent with data obtained in short-term killing assays (**Supplementary Figure 4D**), activation of the A_{2A}R did not significantly modulate killing activity in longer term assays (**Supplementary Figure 4E**). These data are consistent with our conclusions that activation of the A_{2A}R potently suppresses CAR T cell cytokine production but has a more modest effect on *in vitro* cytotoxic activity. Notably, we also provide new data (**New Supplementary Figure 1E**) showing that this suppression of cytokine production can significantly reduce cytokine-mediated killing of tumor cells. Please refer to our response to comment #1 for discussion of this data. Moreover, we agree with the reviewer that our findings in these chromium release assays do not discount the possibility that A_{2A}R suppresses cytotoxic activity *in vivo*, particularly given granzyme B is expressed to a significantly greater extent on A_{2A}R-edited CAR T cells *in vivo* (**Figure 6D**), and we have added additional text to discuss this point on line 421.

“Whilst we observed that A_{2A}R activation suppressed cytokine production by both CD8⁺ and CD4⁺ CAR T cells, we did not observe a role for A_{2A}R in suppressing the direct cytotoxic activity of anti-Lewis Y CAR T cells in these short term assays. However, we cannot discount the possibility that long-term A_{2A}R

activation leads to impaired cytotoxic function *in vivo*, particularly given A_{2A}R-edited CAR T cells displayed higher expression of granzyme B than control CAR T cells when isolated from tumors *ex vivo*.”

7) The antitumour effect of gene edited CAR-T cells in the NSG mouse model is more significant even if it is limited to one single tumor model. However, in this immunodeficient model it remains difficult to assess which is the role of adenosine since there is no tumor microenvironment in these tumor models. The observed antitumor effects may be simply determined by enhanced effector function of gene edited CAR-T cells rather than enhanced persistence at the tumor site which is the main goal of CAR-T cells in solid tumors.

Response: Adenosine is a conserved immunosuppressive pathway in both humans and mice allowing this pathway to be studied in the NSG model since adenosine can be generated by either the OVCAR-3 tumors themselves or myeloid cells and stroma from the host. We have performed additional experiments showing that OVCAR-3 tumor cells express CD73, highlighting their potential to generate adenosine *in vivo*.

Whilst we have not directly measured concentrations of adenosine *in vivo*, as these are complex experiments to undertake, we note that other human ovarian cancer cell lines with similar expression of CD73 have been shown to produce biologically relevant levels of adenosine (PMID 27532024). Therefore, we believe our data indicates that A_{2A}R-editing renders CAR T cells resistant to adenosine locally produced by tumor cells. However, the reviewer is correct in that these effects would likely be even more pronounced with a fully immunocompetent environment as would be expected in a patient, including immunosuppressive subsets such as MDSCs and Tregs that produce high levels of adenosine. Our new data exemplifying the expression of CD73 on OVCAR-3 cells is shown in **New Supplementary Figure 7A** and referred to in the following text on line 311.

“To examine this we treated NSG mice bearing OVCAR-3 tumors, which express high levels of CD73 (**Supplementary Figure 7A**), with A_{2A}R-edited or control anti-Lewis Y CAR T cells.”

To provide increased mechanistic insight in this NSG model we analysed the phenotype of CAR T cells in the spleen and tumors of mice at day 14 post treatment. These experiments revealed that editing of the A_{2A}R significantly enhanced the proportion of intratumoral CAR T cells that expressed IFN γ and TNF, whilst the expression of Ki-67 and the total number of CAR T cells were not significantly enhanced. This data is shown in **New Figures 8E-F** and **Supplementary Figure 7D** and is referred to in the following text on line 319 of the revised manuscript:

“Analysis of intratumoral CAR T cells revealed that CRISPR/Cas9-mediated editing of the A_{2A}R significantly enhanced the proportion of both CD8⁺ and CD4⁺ CAR T cells expressing IFN γ and TNF, whilst the overall numbers of CAR T cells were unaffected (**Figure 8E-F, Supplementary Figure 7D**).”

8) Re-challenge experiments should be performed to assess persistence of CAR-T cells in this model.

Response: To address this question we have performed rechallenge experiments in the Her2 model as we believe that the syngeneic model is superior for analysis of memory responses given that the cytokine support and environmental niche is more clinically relevant. This new data shows that mice previously cured following treatment with A_{2A}R-edited CAR T cells are resistant to a subsequent rechallenge and is shown in **New Supplementary Figure 4G** and is referred to in the following text on line 216:

Postal Address
Locked Bag 1 A'Beckett Street
Victoria 8006 Australia

“Furthermore, mice previously cured following therapy with A_{2A}R-edited CAR T cells were resistant to a secondary challenge with E0771-Her2 on the opposite mammary fat pad, suggesting that A_{2A}R-edited CAR T cells were capable of evoking memory recall responses (**Supplementary Figure 4G**).”

9) Furthermore, a PDX model knowing to recruit murine cells to form the stroma would be helpful.

Response: Whilst we agree that a PDX model with intact stroma would provide additional evidence for the utility of A_{2A}R-edited CAR T cells, we believe that this would constitute an entire separate study.

10) In addition, the control with SCH58261 should be added in these experiments.

Response: Our new data in the NSG model (See response to comment 7 above) shows that A_{2A}R-edited CAR T cells elicit a higher production of IFN γ and TNF when isolated from tumors *ex vivo*. As requested we included an SCH58261 control in this experiment. This data shows that the CRISPR/Cas9 approach is superior in terms of enhancing CAR T cell cytokine production. We have included this data for review only (**Reviewer Only Figure 2**), as we believe this does not significantly strengthen the study given our new data in the murine system (**New Figure 4I**) that conclusively shows that A_{2A}R-edited CAR T cells elicit superior therapeutic activity relative to control CAR T cells combined with SCH58261 treatment.

Reviewer Only Figure 2: A_{2A}R deletion, but not treatment with the A_{2A}R antagonist SCH58261, enhances human CAR T cell function *in vivo*

NSG mice were injected sub-cutaneously with 5×10^6 OVCAR3 tumor cells. Once tumors were established (15-20 mm²) mice were irradiated (1 Gy) and treated with 10×10^6 anti-Lewis Y CAR T cells on two subsequent days. Mice were then treated with 50,000U of IL-2 on days 0-4 post CAR T cell treatment. At day 14 post treatment the proportion of CD8⁺ CAR T cells expressing IFN γ , TNF and Ki-67 was determined. Data shown is the mean \pm SEM of 9 mice per group.

Reviewer 2: This manuscript from Giuffrida and colleagues describes enhancement of CAR-T cell potency in mouse tumor models by CRISPR/Cas9 editing of the adenosine A2AR receptor. They show mechanistic studies indicating that the A2AR-edited cells were resistant to adenosine-mediated transcriptional changes, with increased cytokine production and expression of JAK-STAT signaling pathway genes. These studies are detailed and well done, and make a persuasive case for trials testing the editing of A2AR as a strategy to improve CAR-T function in humans.

Response: We thank the reviewer for their positive assessment of our manuscript.

1) This reviewer strongly feels that the shRNA experiments are dispensable to this paper and that it would benefit from editing for length and to focus on the CRISPR/Cas9 experiments. Further, a more limited version of the shRNA work has been reported twice, by these investigators and another group.

Response: In response to the request from the reviewers 1, 3 and 4 we have performed additional experiments that compare the utility of shRNA and CRISPR/Cas9 approaches and have therefore kept this data in the paper. We hope this is acceptable to this reviewer but we are happy to modify this if requested by the editor. We note that this is the first time that the effects of the shRNA approach have been shown *in vivo*. Importantly, we believe there is significant value to the field in highlighting that effective targeting of the A_{2A}R requires a more stringent strategy of targeting than may be required for other targets where there is less redundancy in the levels of receptor/target expression.

2) Was any effort made to evaluate off-target editing in any of these CAR-T cells? GUIDEseq or similar would be useful to confirm specificity of editing the ADORA2A gene.

Response: We agree these are important studies to perform and have now included this analysis as requested. To address this we used COSMID off target prediction analysis followed by Sanger sequencing of predicted off target sites that were most likely to be targeted in a non-specific fashion by our human A_{2A}R-targeting sgRNA. This analysis revealed undetectable levels of gene editing at these gene loci. This new data is shown in **New Supplementary Figure 6** and referred to in the following text on line 271.

“Moreover, an analysis of potential off target effects induced by this A_{2A}R targeting sgRNA indicated no detectable editing at the only 3 predicted off target sites (Supplementary Figure 6A-B).”

Reviewer #3

1) Adenosine is an immunosuppressive factor that limits anti-tumor immunity through the suppression of multiple immune subsets including T cells via activation of the adenosine A2AR receptor (A2AR). In their previous work that was published in JCI in 2017, the authors showed that either genetic (A2AR-targeted shRNA) or pharmacological targeting of the A2AR profoundly increased CAR T cell efficacy. In this study, they claim that CRISPR/Cas9 mediated gene deletion of A2AR were superior to shRNA mediated knockdown or pharmacological blockade of A2AR. However, the study was separated into two parts, one for shRNA mediated knockdown of A2AR and another one for CRISPR/CAS9 mediated knockdown of A2AR. The conclusions were drawn from comparing two parts of the work. There was no any single experiment that compared shRNA mediated knockdown of A2AR and CRISPR/CAS9 mediated knockdown of A2AR directly, which

Postal Address

Locked Bag 1 A'Beckett Street
Victoria 8006 Australia

is very unusual and the conclusions made from such study design is unacceptable. To make the study valuable with novelty, at least a control group of shRNA mediated knockdown CART has to be included in both Figure 4 and Figure 5.

Response: We have now performed this analysis as requested and agree this adds to the significance of these findings. Please refer to points 3 and 5 in response to Reviewer 1.

Reviewer #4:

1. CAR-T cells generated by CRISPR/Cas9-mediated deletion and shRNA-mediated silencing A_{2A}R should be compared in the same tumor growth experiment along with immunophenotyping in both mouse and human settings.

Response: To address this we have performed a head to head comparison for the effect of CRISPR/Cas9-mediated deletion and shRNA-mediated silencing of A_{2A}R on tumor growth and the immunophenotype of CAR T cells *ex vivo* in the murine setting as suggested. Please refer to comments 3 and 5 in response to reviewer 1. Given the superior efficiency of A_{2A}R targeting by CRISPR/Cas9 in the murine system, we have not optimized an shRNA-based approach for human T cells. However we have provided new data (see response to Reviewer 1 point #10) indicating that CRISPR/Cas9 based targeting of the A_{2A}R is superior to pharmacological targeting with SCH58261 in the human setting.

2. The way CAR-Ts are prepared for the two approaches is different: A_{2A}R is deleted by CRISPR/Cas9 before expansion and in another setting T cells are expanded before silencing. If possible, using similar approaches for both settings will better support the conclusion.

Response: CRISPR/Cas9 editing of activated T cells results in a similar efficiency of knockout to editing of naïve cells. For example the efficiency of Thy1.1 knockout following CRISPR/Cas9 editing is similar when editing naïve or activated T cells (**Reviewer Only Figure 3**). We have previously performed an experiment where anti-Her2 CAR T cells were edited after transduction at day 7. This experiment revealed that CAR T cells that were modified to delete the A_{2A}R elicited significantly greater therapeutic effects relative to control CAR T cells (**Reviewer Figure 4**). This suggests that the timing of CRISPR/Cas9 editing is not critical for rendering cells resistant to adenosine and is consistent with the observation that the addition of a pharmacological antagonist (SCH58261) of the A_{2A}R receptor immediately prior to cultures is also sufficient to reverse NECA-mediated suppression *in vitro*.

We believe that editing of naïve cells is preferable because a) it is more clinically relevant and b) more cost effective as it enables editing of a pool of cells that can be expanded for functional evaluation *in vivo*. We have therefore not included this data in the manuscript but are happy to do so if requested.

Reviewer Only Figure 3- CRISPR/Cas9 mediated deletion of Thy1 is equivalent with naïve or activated murine CAR T cells.

Splenocytes were activated with anti-CD3/anti-CD28 for 16 hours and live T cells obtained by Ficoll centrifugation. CRISPR/Cas9 mediated editing of Thy1 was performed either on freshly isolated T cells (Blue) or after activation with anti-CD3/anti-CD28 (Orange). Cells were then cultured for 7 days in IL-2/ IL-7 containing media and the expression of Thy1 determined.

E0771-Her2 Treatment with anti-Her2 CAR T cells:
CRISPR/Cas9 editing at day 7

Reviewer Only Figure 4 CRISPR/Cas9 mediated deletion of the A₂A_R following activation results in enhanced therapeutic activity.

Anti-Her2 CAR T cells were generated as per Figure 1 of the main manuscript and CRISPR/Cas9 mediated editing of CAR T cells with either an A₂A_R sgRNA or a non-targeting control performed at day 7. Following CRISPR/Cas9 editing, 1×10^7 anti-Her2 CAR T cells were injected intravenously into Her2Tg mice bearing E0771-Her2 tumors as per the standard experimental protocol used in the manuscript. Data is presented as the mean \pm SEM of 6 mice per group.

3. When human CAR-Ts are prepared, cells are activated before the deletion of A₂A_R meaning they may already get A₂A_R signaling along with IL-7/IL-2 signaling until editing with CRISPR/Cas9. In fact, in murine setting CAR-T cells show more effector/exhausted phenotype (more GRZB, PD1 and Tim3, Less CD62L-Figure 5) while that is not the case for human CAR-Ts (Figure 6-7). (Note from author: Due to the inclusion of a new Figure, these comments now refer to Figures 6-8).

Postal Address

Locked Bag 1 A'Beckett Street
Victoria 8006 Australia

Response: Our protocol for editing human CAR T cells after activation is in line with clinically used protocols (PMID 31727131). As discussed above in response to comment 2 we observed that editing of activated murine CAR T cells leads to similar editing efficiency and phenotype but significantly increases costs and so we have not performed extensive repeat experiments to investigate the effect on the *in vivo* phenotype. In both human and murine cells, CRISPR/Cas9 mediated deletion of the A_{2A}R enhances the effector like phenotype of CAR T cells resulting in enhanced production of IFN γ and TNF. Therefore we believe that our results are consistent between models.

4. One major effect of A_{2A}R-mediated immunosuppression is downregulation of IL-2 receptor and IL-2, suggesting the help from CD4 T cells maintaining CD8 T cells in the absence of A_{2A}R is important.

Response: To evaluate the impact of A_{2A}R signaling on the expression of CD25 (IL-2 receptor) and IL-2 we analyzed this in our RNA-Seq experiments. This analysis revealed that CD25 was suppressed by the addition of NECA but we could detect very low levels of IL-2. This new data is shown in **New Supplementary Figure 5H** and is referred to in the following text on line 304.

“These gene lists include several pro-inflammatory cytokines and *IL2RA*, which encodes CD25 the receptor for IL-2 that were sensitive to NECA mediated suppression.”

5. Since the study is very similar to Beavis et al. (PMID: 28165340) one interesting and important addition would be testing the importance of deletion of A_{2A}R in CD4s vs CD8 vs. both to provide a more mechanistic view other than classical *in vitro* cAMP-derived suppression mechanism which may not reflect what is happening *in vivo*.

Response: We thank the reviewer for this suggestion. To address this question we have performed additional experiments showing that adenosine suppresses both CD8⁺ and CD4⁺ CAR T cell function. This data is shown in **New Figure 5A-E** and we have inserted an additional results section on line 220 that describes this new data.

With regards to the impact on *in vivo* efficacy, since our CAR T cell product is 80% CD8⁺ T cells (**Figure 1C**), it was not feasible to generate sufficient numbers of A_{2A}R-edited CD4⁺ CAR T cells for *in vivo* experiments. Therefore to evaluate this question we compared the anti-tumor efficacy of bulk- and CD8⁺ enriched CAR T cells in the context of A_{2A}R editing (or relevant controls). The purity of isolated cell populations is shown in **New Figure 5F** and the anti-tumor response is shown in **New Figure 5G**. This data reveals that CRISPR/Cas9 mediated knockout of A_{2A}R in CD8⁺ CAR T cells is sufficient to enhance their therapeutic efficacy although we note in the discussion section that this does not discount a possible role for enhanced activity of CD4⁺ CAR T cells. This data is referred to in the following text on line 235:

“To assess the relative importance of A_{2A}R deletion in CD8⁺ and CD4⁺ CAR T cells for their *in vivo* efficacy, we purified CD8⁺ CAR T cells from the bulk CAR T cell product following CRISPR/Cas9-editing of the A_{2A}R receptor (**Figure 5F**). The enhanced therapeutic activity of A_{2A}R-edited CAR T cells was observed with either the bulk or CD8⁺ enriched CAR T cell product (**Figure 5G**), indicating

that the deletion of A_{2A}R from CD8⁺ CAR T cells was sufficient to elicit an enhanced therapeutic response.”

We were also able to show through new experiments that CRISPR/Cas9 mediated deletion of the A_{2A}R in human CAR T cells enhances the cytokine production of both CD4⁺ and CD8⁺ CAR T cells *in vivo*. This data is shown in **New Figure 8E-F** and is referred to in the following text on line 319.

“Analysis of intratumoral CAR T cells revealed that CRISPR/Cas9-mediated editing of the A_{2A}R significantly enhanced the proportion of both CD8⁺ and CD4⁺ CAR T cells expressing IFN γ and TNF, whilst the overall numbers of CAR T cells were unaffected (**Figure 8E-F, Supplementary Figure 7D**).”

6. A recent study by Mager et al. suggests that microbiome-derived inosine stimulates A2AR and promotes checkpoint responsiveness. One potential mechanism is A2AR-signaling promoting growth factor responsiveness of T cells. Since CAR-T cells constitutively express CARs they receive CAR-mediated continuous survival signals as opposed to endogenous/antigen-specific conventional T cells, which downregulate their TCRs upon activation and generally do not receive a strong TCR signal. In conventional T cell settings (even in CAR-T setting) all the activation markers presented here is a sign of terminal/effector T cell differentiation and lack of efficacy. This point should be discussed along with proper citation of relevant references (some examples are: PMID: 32792462; PMID: 32792462; PMID: 15980149).

Response: We thank the reviewer for raising this interesting topic. We agree that generating CAR T cells with a more central memory (T_{CM}) phenotype (as elegantly shown in PMID 15980149 referred to by the reviewer) prior to treatment is of high importance for improving the persistence and consequent therapeutic efficacy. Indeed, our recent work (PMID 32735774) highlights this by comparing the efficacy of CAR T cells generated by either IL-15 or IL-2 in the context of the same CAR T cell models used in the current study. Importantly, whilst T_{CM} cells exhibit enhanced persistence, their differentiation into an effector like phenotype is required for anti-tumor activity. Thus, although deletion of the A_{2A}R enhances expression of PD-1 and TIM-3, which can be considered as markers of activation or exhaustion, it also enhances the expression of Granzyme B, Ki-67, IFN γ and TNF. Given the importance of these cytokines/factors in the efficacy of CAR T cells we do not believe this phenotype is consistent with a sign of “lack of efficacy”. Notably, editing of the A_{2A}R does not affect the persistence of CAR T cells or their phenotype in the periphery suggesting that it does not have a negative impact on T cell differentiation in the tumor but rather allows them to effectively differentiate into effector cells within the tumor microenvironment. This is akin to a recent study highlighting that tumors manipulate pathways to promote T cell stemness to suppress anti-tumor immunity (PMID 30923193).

The recent study by Mager *et al.* is of interest and we note that the phenotype observed in this study was predominantly mediated by enhanced T_{H1} differentiation by CD4⁺ T cells. This may not be as important a mechanism in the context of CAR T cells as the cells are expanded *ex vivo* and exhibit a TH1 like phenotype prior to transfer. This could indeed represent an additional benefit of targeting the A_{2A}R by CRISPR/Cas9 as it allows for specific targeting of the A_{2A}R on transferred cells as opposed to pharmacological blockade that targets A_{2A}R on both transferred cells and other endogenous A_{2A}R expressing cells including CD4⁺ T cells. The discussion on this topic is located on line 410.

REVIEWERS' COMMENTS

Reviewer #1 (Remarks to the Author):

This reviewer appreciates the author's effort to improve the manuscript. However, as mentioned in the first revision, since the topic of knocking down A2AR in effector T cells to enhance the antitumor activity and the technology proposed are not particularly novel, the value of the manuscript would rely exclusively on the demonstration of potent antitumour effects. The antitumor effect in immunocompetent mice remains modest. Additionally, the antitumor effects of human CAR-T cells in NSG mice cannot be directly and unequivocally associated with A2AE KO in T cells.

Reviewer #2 (Remarks to the Author):

The authors opted to respond to the other reviewers by adding more shRNA data to meet their criticisms. This was counter to my suggestion that these whole shRNA knock downs approaches are weak and technically ineffective, and Nature's publication space would be better served by focusing on the CRISPR knock out experiments. This is an editorial decision.

The addition of a biased approach to assessing off-target editing (prediction and then limited sequencing of a few genes) lacks rigor and is likely to miss edits that would be revealed by more rigorous and sophisticated unbiased approaches. However, this is not a major point as the gRNA design and editing would have to be very poorly done indeed for off-target edits to account for the observed effects. This is a supplement material-level issue.

Reviewer #3 (not contacted)

Reviewer #4 (report not provided)

Reviewer #5 (Remarks to the Author): to replace Reviewer #4

To the Authors:

This Reviewer was asked whether authors did satisfactory address concerns of Reviewers.

But I could not help, but carefully read this potentially important for cancer therapies Industry paper in general and found some serious and disqualifying scholarship problems, including the factual mistakes.

The Disclaimer to the Editor and Authors:

This Reviewer is VERY protective of this field for obvious reasons. Authors know who is this Reviewer and they should also know that I was originally hesitant to suggest so many of original and first-proof-principle papers from my lab, since it CAN be miss-interpreted as if I am trying to increase my academic credentials at age of 74!

Of course, I don't care about anything like this, having so much appreciation from many scientists, including in earlier papers from these Authors. I am being rewarded beyond my wildest expectations and not only after my IP survived the due diligence by several serious companies, that allowed me to become a Philanthropist. What is the by far most important my reward is the extraordinary joy of learning about the prolonged survivals of those refractory to all other therapies cancer patients, who would otherwise not have any hope, if it were not for our proprietary insights to use the inhibitors of extAdo-generating enzymes, A2AR blockers and anti-hypoxic oxygenation agents.

These drugs that we ushered in are now showing promise in clinical trials in USA hospitals run by prestigious Pharma and the submitted CAR-T approach which is based on our suggestions may yield the best tumor rejections.

My many suggestions are only to better inform the Reader and this –as a bonus- will make a better paper.

First:

Did Authors adequately address Concerns of the Reviewers regarding the experimental part:

1. I am impressed with the attention to details and level of analysis in the appropriately detailed and high-end answers by the Authors, who added some data.

My opinion is that e.g. the Reviewer 4 did ask tough, but fair questions and Authors did give good answers.

Of course, such complicated, expensive, laborious and long-term in vivo cancer immunology projects are never finished 100% and this one is no exception. I must add that in my opinion there is precious few such other academic teams in the world which are capable of performing such study.

But the overall message is sound and after publication this paper will be looked at as the potentially “gold standard” to improve on by others in academia and biotech companies. In this sense paper is ready as is...

2. I recommend to be satisfied with these answers and accept the experimental part of this paper as is, except...

3. Except, I uncovered serious problems with the scholarship in writing this paper and I do strongly recommend to ask for corrections and significant revisions of the Introduction and Discussion, which don't have either appropriately informative Rationale or considerations of limitations of this approach and no warning about the safety issue due to the observations of autoimmunity, which were reported in our foundational - as rated by others-PNAS 2006 paper.

4. It is important to correct these issues and I am sure that it will not be hard in view of detailed suggestions I offered below.

This paper comes from the usually careful, well established and strongest in the field team of Beavis, Darcy- that was originally put together by great scientist Mark Smyth and I have strong reasons to expect that this paper will be scrutinized by the high power immunologists and SAB members of the top I-O companies, which are also developing this and/or similar approach with CARs, where Beavis, Darcy are the Leaders due to their previous papers.

5. Many experts will analyze this paper in Nature Comm. in order to approve or deny the significant investments into clinical trials of A2AR-/- CAR-T cells.

From my own experience in serving as an expert, these are the major questions they will have and the revised paper should have good answers:

A) How solid is the foundational scientific justification of this project of targeting/removing/blocking the A2AR in anti-tumor T cells?

B) Were these foundational studies done in high reputation labs and by serious scientists?

C) What are the mechanisms of expected to be stronger anti-tumor effects of A2AR gene- deleted i CAR-T cells?

D) What did authors say about the potential safety issues and limitations of their approach in humans and what is the suggested roadmap to future improvement?

The answers to these questions either not presented at all or not clearly presented in this to be revised paper.

The inadequate and haphazard Rationale in Introduction and lack of discussion of limitations would scientifically negate the overall impact of this paper on the interested Readers of Nature Communications journal.

However, Authors should not be too upset since it will be easy to correct.

It is important for me that Authors feel that this Reviewer-while being critical- is having the best possible attitude. This is because of their Mentor Mark Smyth' vision to
i) join the lonely, but persistent , ~20 years -long efforts of my Team and
ii) his ability to recruit and persuade the best Members of his Team - like Stagg, Beavis and Darcy. This is why the world now has much better chance to have the curative immunotherapies of cancer.

I believe that Beavis and Darcy may be having the best chances to translate the CAR-T into successful rejections of solid tumors.

Criticisms and Recommendations

As I mentioned , this paper will be scrutinized by high power immunologists and SAB members of the top I-O companies and I also know that the majority of these SAB members and Readers of Nature Communications are NOT versed in the background of anti-Hypoxia-A2-adenosinergic immunotherapies and this is why this must be much better explained.

This why I did evaluate the each reference from the point of view of the sophisticated Reader, who wants to go to the SOURCE with the most details and insights.

It is expected from serious journals, like Nature Communications, that serious Authors will identify such original publications with the most insights for the Reader.

Suggestions:

To correct factual mistakes and provide much better scientific justification of removing A2AR-from T cells, including what was done before who and how did that and what is the mechanism of anti-tumor effects of A2AR gene deletion?

The Reader of Nature Communications will benefit by better understanding of the biochemical and immunological background behind the Rationale

This, in turn, requires to revise the sequence of messages in the Introduction and provide the references papers in the correct order of the foundational significance of messages in these papers. This will will require mostly to rearrange the already referred to papers.

1st Set of Suggestions:

To provide the factual and sound scientific justification of removing A2aR in CAR-T cells in the Introduction of the paper in order to better explain the biochemical and immunological background behind the Rationale.

1A-

To describe what are the foundational observations that authors later confirm and advanced by their studies of CAR-T cells?

-What is the mechanism of the anti-tumor effects of A2AR gene deletion that the CAR-T cells will re-acquire after the deletion of the A2AR?

Here are the most important insights of Ohta et al in Nature and PNAS that authors should include in the very beginning so that Reader will be informed.

Authors should re-write the paper with the understating that it is impossible to build the First floor and Second floor without the foundation. This was well understood by their Mentor Mark Smyth and reflected in the subsequent papers from his team, including these Authors:

The Foundation is based on Nature 2001 and PNAS 2006 papers: that not only created the novel academic field, but it also made it obvious how to further improve the clinical outcomes of immunotherapies by :

- i) therapeutically eliminate the extAdo-generating enzymes-including the CD39/CD73;
- ii) therapeutically eliminate the Hypoxia-HIF-1alpha upstream;
- iii) therapeutically block the A2AR with antagonist and-for a good measure:-
- iv) to use the ADA deaminase to decrease the extAdo in cancer tissues

Below are the major scientific messages that the Reader SHOULD be informed about in the beginning of the paper on CAR-T cells.

- 1. The extAdo-A2AR-cAMP axis is critically important in controlling immune response and inflammatory tissue damage and anti-tumor immunity
 - =Ref on: Ohta A, Sitkovsky M Role of G-protein-coupled adenosine receptors in downregulation of inflammation and protection from tissue damage. Nature. 2001 Dec 20-27;414(6866):916-20.
 - =Ref on Ohta at all PNAS 2006
- 2. The extAdo-A2AR-cAMP axis is also NON-redundant, since it was specifically demonstrated that other cAMP-elevating GsPCRs will not compensate for A2AR
 - Ref= Nature. 2001
- 3. The proof of principle for the conceptually novel immunotherapies of cancer and infectious diseases was provided by showing that
 - i) genetic deletion or
 - ii) pharmacological blockade of the cAMP-elevating A2AR leads to the powerful unleashing of the anti-pathogen or anti-tumor effector functions of immune cells and causes the rejection of established tumors-Nature. 2001 and PNAS 2006
- 4. Why CAR-T without A2AR are expected to be effective?

-The Mechanism of A2AR-blockade or gene deletion-enabled killing of tumor cells was suggested by Ohta et al as to be due to increase in level of secretion by anti-tumor T cells of the IFNgamma.

The high IFNgamma then causes the "starvation" of tumors.

"Starvation" BECAUSE according to Tony Blankenstein (Immunity, 1997?) the higher levels of the CTL-generated IFNgamma, that are due to the blocked or genetically eliminated A2AR- will prevent the tumor-saving feeding and oxygenating neo-vascularization.

****It should be explained to the Reader that A2AR blockade/deletion is so powerful NOT because it is moderately de-inhibiting the direct perforin-granzyme killing, but because it unleashes the IFN-gamma-mediated inhibition of neovascularization and thereby STARVES tumors (Ohta at all. PNAS 2006)..

- 5. Authors may also address the potential concern about their approach of targeting ONLY the high affinity A2AR, but not the low affinity A2BR by referring to papers, that provided the evidence that the immunosuppressive and cAMP-elevating A2BR does not compensate for A2AR:
 - -Ref Lukashev DE, Smith PT, Caldwell CC, Ohta A, Apasov SG, Sitkovsky MV Analysis of A2a receptor-deficient mice reveals no significant compensatory increases in the expression of A2b, A1, and A3 adenosine receptors in lymphoid organs. Biochem Pharmacol. 2003 Jun 15;65(12):2081-90.

- Also mention that A2BR can NOT compensate for A2AR: because the A2AR is needed for A2BR expression on lymphocytes.
 -
 - =Ref on Moriyama K, Sitkovsky MV Adenosine A2A receptor is involved in cell surface expression of A2B receptor. . J Biol Chem. 2010 Dec 10;285(50):39271-88.
 -
6. Earlier use of siRNA to target the A2AR on anti-tumor T cells which is of the immediate relevance to the submitted genetic CRISPR approach:

The PNAS 2006 paper provided the first evidence that genetic deletion of A2AR in the to be adoptively transferred T cells by siRNA unleashed these anti-tumor T cells to kill tumors

=Ref on Ohta A, Gorelik E, Prasad SJ, Ronchese F, Lukashev D, Wong MK, Huang X, Caldwell S, Liu K, Smith P, Chen JF, Jackson EK, Apasov S, Abrams S, Sitkovsky M. A2A adenosine receptor protects tumors from antitumor T cells. Proc Natl Acad Sci U S A. 2006 Aug 29;103(35):13132-7.

7. Authors should specifically emphasize that it is VERY important for their approach that effects of A2AR deletion on improvement of tumor rejection are T-CELL-AUTONOMOUS and that this was prominently messaged in the Abstract PNAS 2006 by Ohta at all.

"The data suggest that effects of A2AR are T cell autonomous"

These data provided the mechanistic basis to expect that A2AR-mediated inhibition is T-CELL-AUTONOMOUS, thereby justifying the elimination of A2AR only in CAR-T killer cells.

In order to test and prove this Ohta at all made a multi-directional effort and one of cancer immunology designs that they employed was the targeting the A2A receptors on CTL by siRNA pre-treatment. CTL were then adoptively transferred into tumor-growing mice.

****FYI:

Authors may ask how was it possible to test so much in that one PNAS 2006 paper?

-They should know that that PNAS paper was in production for ~6 years and it was a huge undertaking of the international Consortium of NIH, NIAID and NIH, NCI and Pitt USA, and Malaghan Inst, New Zealand.

2nd Set of Suggestions:

My advice is to revise the beginning of the Introduction with which is something like:

1 =CAR-T cell were shown to be effective in rejecting tumors in lymphoma patients, but CAR-T failed to reject solid tumors, prompting US-i.e. Beavis/Darcy- to test and provide the proof of principle for the therapeutic promise of the blockade of A2AR in preclinical studies.

Therefore the ref#1 in this paper should be ref on the earlier CAR-T cells paper from these Authors as the proof of principle for this specific paper.

=Beavis, P.A. et al. Targeting the adenosine 2A receptor enhances 827 chimeric antigen receptor T cell efficacy. J Clin Invest 127, 929-941 (2017).

=Then I suggest to proceed with providing the first and strongest justifications to genetically eliminate the cAMP-elevating A2AR in adoptively transferred CTL as is the Aim of submitted paper.

- Therefore the ref#2,3 should be on Ohta et al Nature 2001 and 2006 PNAS papers which created this Anti-A2-adenosinergic field of cancer immunology.

=Ref 2 Ohta A, Sitkovsky M Role of G-protein-coupled adenosine receptors in downregulation of inflammation and protection from tissue damage. *Nature*. 2001 Dec 20-27;414(6866):916-20.

Ref 3 Ohta A, Gorelik E, Prasad SJ, Ronchese F, Lukashev D, Wong MK, Huang X, Caldwell S, Liu K, Smith P, Chen JF, Jackson EK, Apasov S, Abrams S, Sitkovsky M. A2A adenosine receptor protects tumors from antitumor T cells. *Proc Natl Acad Sci U S A*. 2006 Aug 29;103(35):13132-7.

3rd Set of Suggestions:

To improve the Discussion by explaining the possible limitations of the proposed approach and what to be alert about in terms of safety and efficacy:

1. To provide references on the most important Reviews-Opinions, not just Reviews with collection of factoids...

I recommend the

=Vijayan D, Young A, Teng MWL, Smyth MJ. Targeting immunosuppressive adenosine in cancer. *Nat Rev Cancer*. 2017 Dec;17(12):709-724. Review.

=*Cancer Discov*. 2020 Jan;10(1):16-19. Lessons from the A2A Adenosine Receptor Antagonist-Enabled Tumor Regression and Survival in Patients with Treatment-Refractory Renal Cell Cancer. Sitkovsky MV.

= The "Hostile TME..." In *Cancer Immunology Research* by Sitkovsky and Hatfield...

3-1

Important Safety Warning:

Authors should discuss the issue of autoimmunity due to the A2AR deletion in T cells, which was first actually noticed and emphasized in PNAS 2006 paper including the possible solution by restarting this A2AR pathway...

From the Abstract of PNAS 2006 paper:

- The observations of autoimmunity during melanoma rejection in A2AR-deficient mice suggest that A2AR in T cells is also important in preventing autoimmunity. Thus, although using the hypoxiaadenosineA2AR pathway inhibitors may improve antitumor immunity, the recruitment of this pathway by selective drugs is expected to attenuate the autoimmune tissue damage.

3-2

To discuss that even the A2AR deficient CARs could be STILL inhibited by other immunosuppressive molecules

-like TGF-beta- and T regs and MDSC in the most resistant to inhibition hypoxia-HIF and extAdo—A2AR∆cAMP-rich TMEs.

This is due to the "hypoxia-HIF-1alpha-HRE mediated and extAdo—A2AR-cAMP-mediated immunosuppressive transcription"

-which was the conceptually novel concept when it was put forward 11 years ago by Sitkovsky MV in T regulatory cells: hypoxia-adenosinergic suppression and re-direction of the immune response. *Trends Immunol.Cell*, 2009 Mar;30(3):102-8.

3-3

Authors should address the troubling possibility of compensation for immunosuppressive A2AR by also immunosuppressive , cAMP-elevating A2BR in their CAR-T cells.

If likely, this would defeat the purpose of depleting the CAR-T-cells from the A2AR.

Authors may stronger justify their approach by referring to the experimental evidence that the immunosuppressive and cAMP-elevating A2BR adenosine receptor does not compensate for A2AR:

Ref Lukashev DE, Smith PT, Caldwell CC, Ohta A, Apasov SG, Sitkovsky MV Analysis of A2a receptor-deficient mice reveals no significant compensatory increases in the expression of A2b, A1, and A3 adenosine receptors in lymphoid organs. *Biochem Pharmacol.* 2003 Jun 15;65(12):2081-90.

It was reported in this paper, that

"These data suggest that regulation of the expression of A2b, A1, and A3 receptors is not affected significantly by the absence of A2a receptors and may provide further explanation of earlier in vivo observations of increased tissue damage and of longer persistence of proinflammatory cytokines in animals with inactivated A2a receptors".

3-4

Mention that the likely reason that the immunosuppressive A2BR can not compensate for A2AR is because the A2AR is required for the A2BR expression on the surface of lymphocytes.

- =Ref on Moriyama K, Sitkovsky MV Adenosine A2A receptor is involved in cell surface expression of A2B receptor. *J Biol Chem.* 2010 Dec 10;285(50):39271-88.

-Thus, Authors should emphasize that such highly undesirable compensatory upregulation of immunosuppressive A2BR in their CAR-T with the deletion of A2AR.

4th Set of Suggestions:

What is the roadmap to future improvement?

Authors may offer the therapeutic prevention of the still strong inhibition of A2AR-deficient CAR-T cells in the hypoxic and adenosine-rich TME by OTHER immunosuppressors that are triggered by hypoxia-HIF-1 and adenosine-A2AR-cAMP-mediated immunosuppressive transcription, that could be much more influential in "old" human tumors than in 2-3 week old mouse tumors

Authors can find these therapeutic prevention measures recently described in the invited Opinions and Reviews below:

1. Sitkovsky MV.

Lessons from the A2A Adenosine Receptor Antagonist-Enabled Tumor Regression and Survival in Patients with Treatment-Refractory Renal Cell Cancer. *Cancer Discov.* 2020 Jan;10(1):16-19.

and these measures include the STOP of the Hypoxia-A2A adenosinergic pathway upstream by eliminating the TME hypoxia

2. Hatfield SM, Sitkovsky MV.

Antihypoxic oxygenation agents with respiratory hyperoxia to improve cancer immunotherapy. *J Clin Invest.* 2020 Nov 2;130(11):5629-5637.

5th Set of Suggestions:

5-1

Authors should improve the statement below by referencing not only their own 2015/2017 studies but also much earlier and careful 1997 paper...

"A major immunosuppressive phenotype mediated by A2AR activation is impaired cytokine production by T cells whilst cytotoxic activity is less affected^{16, 29}.

Ref 16= Beavis, P.A. et al. Targeting the adenosine 2A receptor enhances 827 chimeric antigen receptor T

cell efficacy. *J Clin Invest* 127, 929-941 (2017).

Ref 29 Beavis, P.A. et al. Adenosine Receptor 2A Blockade Increases the Efficacy of Anti-PD-1 868 through Enhanced Antitumor T-cell Responses. *Cancer Immunol Res* 3, 506-517 (2015).

-The correct reference that started these comparisons is in

a. Koshiba M, Kojima H, Huang S, Apasov S, Sitkovsky MV.. Memory of extracellular adenosine A2A purinergic receptor-mediated signaling in murine T cells. *J Biol Chem*. 1997 Oct 10;272(41):25881-9. It was shown that extracellular Ado (extAdo) suppresses all tested T cell receptor (TCR)-triggered effector functions of T lymphocytes including the stronger inhibition of the TCR-triggered FasL mRNA up-regulation in cytotoxic T lymphocytes.

b.

Cutting edge: Physiologic attenuation of proinflammatory transcription by the Gs protein-coupled A2A adenosine receptor in vivo.

Lukashev D, Ohta A, Apasov S, Chen JF, Sitkovsky M. *J Immunol*. 2004 Jul 1;173(1):21-4. doi:

c. And very relevant reference that was first to show in vivo in tumor rejections assays that "A2AR/A2BR Antagonists May Dehibit the Production of IFN- γ by Antitumor T Cells in the Tumor Microenvironment".

Ohta A, Gorelik E, Prasad SJ, Ronchese F, Lukashev D, Wong MK, Huang X, Caldwell S, Liu K, Smith P, Chen JF, Jackson EK, Apasov S, Abrams S, Sitkovsky M.

A2A adenosine receptor protects tumors from antitumor T cells. *Proc Natl Acad Sci U S A*. 2006 Aug 29;103(35):13132-7.

5-2

Authors must change this only partially correct statement:

As is , Authors write:

"Two previous studies, including our own, have characterized the effect of shRNA in modulating A2AR mediated suppression

References =16, 42

but these studies were limited to in vitro assessment of cytokine production and killing capacity.

16-Beavis, P.A. et al. Targeting the adenosine 2A receptor enhances chimeric antigen receptor T cell efficacy. *J Clin Invest* 127, 929-941 (2017)

42-Masoumi, E. et al. Genetic and pharmacological targeting of A2a receptor improves function of anti-mesothelin CAR T cells. *J Exp Clin Cancer Res* 39, 49 (2020).

In fact, the first, original justifications to genetically eliminate A2AR in adoptively transferred CTL that is the main aim of submitted study were presented in Ohta et al. PNAS

Therefore, describing the effect of shRNA Authors should reference the PNAS 2006 paper by Ohta et al:

In that paper the genetic, siRNA targeted knock-down of A2AR by pretreatment of anti-tumor T cells before their adoptive transfer into tumor-growing mice was shown to result in better anti-tumor immunity in vivo and that was interpreted as one of the several presented lines of evidence, that effects of A2AR deletion are T cell-autonomous.

Ref Ohta A, Gorelik E, Prasad SJ, Ronchese F, Lukashev D, Wong MK, Huang X, Caldwell S, Liu K,

Smith P, Chen JF, Jackson EK, Apasov S, Abrams S, Sitkovsky M. and
A2A adenosine receptor protects tumors from antitumor T cells. Proc Natl Acad Sci U S A. 2006
Aug 29;103(35):13132-7.

5-3

Coming back to the i) importance of the evidence that effects of A2AR deletion on improvement of tumor rejection are T-CELL-AUTONOMOUS.

This was prominently messaged in the Abstract PNAS 2006 by Ohta at all. and then in
Cekic C, Linden J. paper " A2A receptors intrinsically regulate CD8 T cells in their title Adenosine
A2A receptors intrinsically regulate CD8+ T cells in the tumor microenvironment. Cekic C, Linden
J. Cancer Res. 2014 Dec 15;74(24):7239-49,

As a Reviewer with 38 years of publishing in top journals immunological studies of the T cell
selection in thymus, TCR- activation, differentiation and effector functions,
I must alert Authors about weaknesses in papers they refer to in order to protect and not to
mislead the Reader.

This is why I recommend Authors to critically re-visit the paper from Cecic-Linden, before
referencing it in the prestigious Nature Communications..

In my view this paper has several disqualifying mistakes in design and interpretations and in
scholarship. I am very surprised that Authors of that paper succeeded in persuading their
Reviewers and the Editor of Cancer Res. that their message is really novel (!?), so that they were
even allowed to have it as title!

- even that their paper was 8 years later after the PNAS 2006 paper where this same message was
prominently displayed even in the Abstract.

I think they managed to persuade their Reviewers and the Editor of Cancer Res. that their
message is really novel by employing a clever "word play", so that instead of "T cell
AUTONOMOUS" wording that Ohta at all was prominently messaging 8 years earlier they employed
the different, wording i.e. the synonymous " A2A receptors INTRINSICALLY regulate CD8 T cells.

While this word play on itself is not sufficient to avoid referencing this paper, the poor
experimental design and the over-interpretation of their observations are indeed, serious
disqualifying problems.

This is why it is "Poor experimental design" that leads to artifacts;

-Linden's paper was done with T cell specific A2AR-knock-down mice, which are the
"immunological freaks of nature" in terms of differentiation, and memory T cells due to the
screwed up negative and positive selection in thymus of the A2AR-/- thymocytes.

The TCR-repertoire and subsequent T cells memory "signals" are ALSO screwed up. -As a result,
the behavior of the resulting peripheral T cells will be also abnormal and of relevance of only to
this A2AR-knock-down mice.

Indeed, memory T cells processes are heavily dependent on the TCR signaling...

There is only so much you can accept in terms of "hints" from such highly artificial mice without
making special pharmacological controls or what Beavis and Darcy did in their papers.

I accept only the small part of observations trustworthy in Cecic-Linden paper and only because
they are in agreement with earlier published controls by other good scientists.

This is why Authors of CAR-T paper do not need to "validate" in discussion chapter their strong and
straightforward observations with the weak data in the -what I consider inferiorly designed and
over-interpreted experiments and predictions.

##It must be noticed that the negative predictions of in Cecic-Linden paper about using A2AR
blockade method, were NOT confirmed in subsequent clinical trials of A2AR blockade.

FYI,
according to the imaging radiology controls in humans , the blockade with A2AR blocker from Vernalis in Corvus trial is 100% of A2AR binding sites, so this is actually the "Pharmacological KO" of A2AR in humans.

Accordingly, the foundational Ohta et al, PNAS 2006 paper was both about i) A2AR KO and ii) about the pharmacological KO of A2AR in mouse WT anti-tumor T cells, that were not irreversibly screwed up.

Reviewer
Michail Sitkovsky

Response to Reviewer Comments

Reviewer #1 (Remarks to the Author):

This reviewer appreciates the author's effort to improve the manuscript. However, as mentioned in the first revision, since the topic of knocking down A2AR in effector T cells to enhance the antitumor activity and the technology proposed are not particularly novel, the value of the manuscript would rely exclusively on the demonstration of potent antitumour effects. The antitumor effect in immunocompetent mice remains modest. Additionally, the antitumor effects of human CAR-T cells in NSG mice cannot be directly and unequivocally associated with A2AE KO in T cells.

Response: We are glad the reviewer appreciates our efforts to improve the manuscript. We respectfully disagree with the reviewer's assertions on novelty and significance. We believe our paper is novel since CRISPR/Cas9 deletion of the A_{2A}R has never been described before and is of high clinical significance since our data indicates that a CRISPR/Cas9 strategy is superior to pharmacological targeting of the receptor. Our data targeting the A_{2A}R using CRISPR/Cas9 shows high potential for improving CAR T cell therapy for patients particularly against solid cancers particularly given the fact that this technology has already been utilized in the clinic.

Reviewer #2 (Remarks to the Author):

The authors opted to respond to the other reviewers by adding more shRNA data to meet their criticisms. This was counter to my suggestion that these whole shRNA knock downs approaches are weak and technically ineffective, and Nature's publication space would be better served by focusing on the CRISPR knock out experiments. This is an editorial decision.

The addition of a biased approach to assessing off-target editing (prediction and then limited sequencing of a few genes) lacks rigor and is likely to miss edits that would be revealed by more rigorous and sophisticated unbiased approaches. However, this is not a major point as the gRNA design and editing would have to be very poorly done indeed for off-target edits to account for the observed effects. This is a supplement material-level issue.

Response: We thank the reviewer for the comment. However, we respectfully disagree that our approach to assess off target effects lacks rigor since we analyzed all 3 off target sites predicted by an unbiased algorithm-based software (COSMID). Notably this tool is commonly used in the field and is broadly accepted as an appropriate means of identifying off target sites (PMID 32241852). We agree that a broader approach such as CRISPRSeq or GUIDESeq would further alleviate these concerns, and could be implemented prior to clinical translation.

Reviewer #3 (not contacted)

Reviewer #4 (report not provided)

Reviewer #5 (Remarks to the Author): to replace Reviewer #4

To the Authors:

This Reviewer was asked whether authors did satisfactory address concerns of Reviewers.

But I could not help, but carefully read this potentially important for cancer therapies Industry paper in general and found some serious and disqualifying scholarship problems, including the factual mistakes.

The Disclaimer to the Editor and Authors:

This Reviewer is VERY protective of this field for obvious reasons. Authors know who is this Reviewer and they should also know that I was originally hesitant to suggest so many of original and first-proof-principle papers from my lab, since it CAN be miss-interpreted as if I am trying to increase my academic credentials at age of 74!

Of course, I don't care about anything like this, having so much appreciation from many scientists, including in earlier papers from these Authors. I am being rewarded beyond my wildest expectations and not only after my IP survived the due diligence by several serious companies, that allowed me to become a Philanthropist. What is the by far most important my reward is the extraordinary joy of learning about the prolonged survivals of those refractory to all other therapies cancer patients, who would otherwise not have any hope, if it were not for our proprietary insights to use the inhibitors of extAdo-generating enzymes, A2AR blockers and anti-hypoxic oxygenation agents.

These drugs that we ushered in are now showing promise in clinical trials in USA hospitals run by prestigious Pharma and the submitted CAR-T approach which is based on our suggestions may yield the best tumor rejections.

My many suggestions are only to better inform the Reader and this as a bonus- will make a better paper.

First:

Did Authors adequately address Concerns of the Reviewers regarding the experimental part:

1. I am impressed with the attention to details and level of analysis in the appropriately detailed and high-end answers by the Authors, who added some data.

My opinion is that e.g. the Reviewer 4 did ask tough, but fair questions and Authors did give good answers.

Of course, such complicated, expensive, laborious and long-term in vivo cancer immunology projects are never finished 100% and this one is no exception. I must add that in my opinion there is precious few such other academic teams in the world which are capable of performing such study.

But the overall message is sound and after publication this paper will be looked at as the potentially gold standard to improve on by others in academia and biotech companies.

In this sense paper is ready as is???

2. I recommend to be satisfied with these answers and accept the experimental part of this paper as is, except???

3. Except, I uncovered serious problems with the scholarship in writing this paper and I do strongly recommend to ask for corrections and significant revisions of the Introduction and Discussion, which don't have either appropriately informative Rationale or considerations of limitations of this approach and no warning about the safety issue due to the observations of autoimmunity, which were reported in our foundational - as rated by others-PNAS 2006 paper.

4. It is important to correct these issues and I am sure that it will not be hard in view of detailed suggestions I offered below.

This paper comes from the usually careful, well established and strongest in the field team of Beavis, Darcy- that was originally put together by great scientist Mark Smyth and I have strong reasons to expect that this paper will be scrutinized by the high power immunologists and SAB members of the top I-O companies, which are also developing this and/or similar approach with CARs, where Beavis, Darcy are the Leaders due to their previous papers.

5. Many experts will analyze this paper in Nature Comm. in order to approve or deny the significant investments into clinical trials of A2AR-/- CAR-T cells.

From my own experience in serving as an expert, these are the major questions they will have and the revised paper should have good answers:

A) How solid is the foundational scientific justification of this project of targeting/removing/blocking the A2AR in anti-tumor T cells?

B) Were these foundational studies done in high reputation labs and by serious scientists?

C) What are the mechanisms of expected to be stronger anti-tumor effects of A2AR gene- deleted i CAR-T cells?

D) What did authors say about the potential safety issues and limitations of their approach in humans and what is the suggested roadmap to future improvement?

The answers to these questions either not presented at all or not clearly presented in this to be revised paper.

The inadequate and haphazard Rationale in Introduction and lack of discussion of limitations would scientifically negate the overall impact of this paper on the interested Readers of Nature Communications journal.

However, Authors should not be too upset since it will be easy to correct.

It is important for me that Authors feel that this Reviewer-while being critical- is having the best possible attitude. This is because of their Mentor Mark Smyth's vision to

i) join the lonely, but persistent, ~20 years -long efforts of my Team and

ii) his ability to recruit and persuade the best Members of his Team - like Stagg, Beavis and Darcy.

This is why the world now has much better chance to have the curative immunotherapies of cancer.

I believe that Beavis and Darcy may be having the best chances to translate the CAR-T into successful rejections of solid tumors.

Criticisms and Recommendations

As I mentioned, this paper will be scrutinized by high power immunologists and SAB members of the top I-O companies and I also know that the majority of these SAB members and Readers of Nature Communications are NOT versed in the background of anti-Hypoxia-A2-adenosinergic immunotherapies and this is why this must be much better explained.

This why I did evaluate the each reference from the point of view of the sophisticated Reader, who wants to go to the SOURCE with the most details and insights.

It is expected from serious journals, like Nature Communications, that serious Authors will identify such original publications with the most insights for the Reader.

Suggestions:

To correct factual mistakes and provide much better scientific justification of removing A2AR-from T cells, including what was done before who and how did that and what is the mechanism of anti-tumor effects of A2AR gene deletion?

The Reader of Nature Communications will benefit by better understanding of the biochemical and immunological background behind the Rationale

This, in turn, requires to revise the sequence of messages in the Introduction and provide the references papers in the correct order of the foundational significance of messages in these papers. This will require mostly to rearrange the already referred to papers.

1st Set of Suggestions:

To provide the factual and sound scientific justification of removing A2aR in CAR-T cells in the Introduction of the paper in order to better explain the biochemical and immunological background behind the Rationale.

1A-

To describe what are the foundational observations that authors later confirm and advanced by their studies of CAR-T cells?

-What is the mechanism of the anti-tumor effects of A2AR gene deletion that the CAR-T cells will acquire after the deletion of the A2AR?

Here are the most important insights of Ohta et al in Nature and PNAS that authors should include in the very beginning so that Reader will be informed.

Authors should re-write the paper with the understating that it is impossible to build the First floor and Second floor without the foundation. This was well understood by their Mentor Mark Smyth and reflected in the subsequent papers from his team, including these Authors:

The Foundation is based on Nature 2001 and PNAS 2006 papers:
that not only created the novel academic field, but it also made it obvious how to further improve the clinical outcomes of immunotherapies by :

- i) therapeutically eliminate the extAdo-generating enzymes-including the CD39/CD73;
- ii) therapeutically eliminate the Hypoxia-HIF-1alpha upstream;
- iii) therapeutically block the A2AR with antagonist and-for a good measure:-
- iv) to use the ADA deaminase to decrease the extAdo in cancer tissues

Below are the major scientific messages that the Reader SHOULD be informed about in the beginning of the paper on CAR-T cells.

??? 1.The extAdo-A2AR-cAMP axis is critically important in controlling immune response and inflammatory tissue damage and anti-tumor immunity

??? =Ref on: Ohta A, Sitkovsky M Role of G-protein-coupled adenosine receptors in downregulation of inflammation and protection from tissue damage. Nature. 2001 Dec 20-27;414(6866):916-20.

??? =Ref on Ohta at all PNAS 2006

??? 2.The extAdo-A2AR-cAMP axis is also NON-redundant, since it was specifically demonstrated that other cAMP-elevating GsPCRs will not compensate for A2AR

??? Ref= Nature. 2001

??? 3.The proof of principle for the conceptually novel immunotherapies of cancer and infectious diseases was provided by showing that

i) genetic deletion or

ii) pharmacological blockade of the cAMP-elevating A2AR leads to the powerful unleashing of the anti-pathogen or anti-tumor effector functions of immune cells and causes the rejection of established tumors -Nature. 2001 and PNAS 2006

??? 4. Why CAR-T without A2AR are expected to be effective?

-The Mechanism of A2AR-blockade or gene deletion-enabled killing of tumor cells was suggested by Ohta et al as to be due to increase in level of secretion by anti-tumor T cells of the IFNgamma.

The high IFNgamma then causes the ???starvation??? of tumors.

"Starvation" BECAUSE according to Tony Blankenstein (Immunity, 1997?) the higher levels of the CTL-generated IFNgamma,

that are due to the blocked or genetically eliminated A2AR- will prevent the tumor-saving feeding and oxygenating neo-vascularization.

****It should be explained to the Reader that A2AR blockade/deletion is so powerful NOT because it is moderately de-inhibiting the direct perforin-granzyme killing,
but because it unleashes the IFN-gamma-mediated inhibition of neovascularization and thereby STARVES tumors (Ohta at all. PNAS 2006)..

??? 5. Authors may also address the potential concern about their approach of targeting ONLY the high affinity A2AR, but not the low affinity A2BR by referring to papers, that provided the evidence that the immunosuppressive and cAMP-elevating A2BR does not compensate for A2AR:

???

??? -Ref Lukashev DE, Smith PT, Caldwell CC, Ohta A, Apasov SG, Sitkovsky MV Analysis of A2a receptor-deficient mice reveals no significant compensatory increases in the expression of A2b, A1, and A3 adenosine receptors in lymphoid organs. *Biochem Pharmacol.* 2003 Jun 15;65(12):2081-90.

??? Also mention that A2BR can NOT compensate for A2AR: because the A2AR is needed for A2BR expression on lymphocytes.

???

??? =Ref on Moriyama K, Sitkovsky MV Adenosine A2A receptor is involved in cell surface expression of A2B receptor. *J Biol Chem.* 2010 Dec 10;285(50):39271-88.

???

6. Earlier use of siRNA to target the A2AR on anti-tumor T cells which is of the immediate relevance to the submitted genetic CRISPR approach:

The PNAS 2006 paper provided the first evidence that genetic deletion of A2AR in the to be adoptively transferred T cells by siRNA unleashed these anti-tumor T cells to kill tumors

=Ref on Ohta A, Gorelik E, Prasad SJ, Ronchese F, Lukashev D, Wong MK, Huang X, Caldwell S, Liu K, Smith P, Chen JF, Jackson EK, Apasov S, Abrams S, Sitkovsky M. A2A adenosine receptor protects tumors from antitumor T cells. *Proc Natl Acad Sci U S A.* 2006 Aug 29;103(35):13132-7.

7. Authors should specifically emphasize that it is VERY important for their approach that effects of A2AR deletion on improvement of tumor rejection are T-CELL-AUTONOMOUS and that this was prominently messaged in the Abstract PNAS 2006 by Ohta at all.

???The data suggest that effects of A2AR are T cell autonomous???

These data provided the mechanistic basis to expect that A2AR-mediated inhibition is T-CELL-AUTONOMOUS, thereby justifying the elimination of A2AR only in CAR-T killer cells.

In order to test and prove this Ohta at all made a multi-directional effort and one of cancer immunology designs that they employed was the targeting the A2A receptors on CTL by siRNA pre-treatment. CTL were then adoptively transferred into tumor-growing mice.

****FYI:

Authors may ask how was it possible to test so much in that one PNAS 2006 paper?

-They should know that that PNAS paper was in production for ~6 years and it was a huge undertaking of the international Consortium of NIH, NIAID and NIH, NCI and Pitt USA, and Malaghan Inst, New Zealand.

Response: We thank the reviewer for their suggestions as a pioneer in the adenosine field. Our introduction aims to provide context to the reader from the perspective of both the CAR T cell and adenosine fields. We therefore apologize if the rationale was not clearly presented to the reviewer with

specific reference to the adenosine field. To further improve this we have inserted the following passages of text to the reader.

Line 41: “Formative studies by the group of Sitkovsky identified that the extracellular adenosine-A_{2A}R-cAMP axis is critically important in controlling immune responses, inflammatory tissue damage and anti-tumor immunity^{6,7}.”

Line 47: “A_{2A}R deficient mice elicit significantly enhanced T cell responses and notably this A_{2A}R-cAMP axis is non-redundant, since other cAMP-elevating G-Protein Coupled Receptors do not compensate for the loss of A_{2A}R⁶.”

Line 53: “Furthermore this work has set the scene for alternative strategies to target this pathway including therapeutics directed towards the upstream ectoenzymes CD73, CD39 and CD38, or the upstream Hypoxia-HIF-1 α axis itself^{8,10} signifying the clinical interest in targeting this pathway¹³.”

In addition we appreciate the reviewer’s comments on the intrinsic link between A_{2A}R-mediated suppression and IFN γ production. Our data fully supports this model and the reviewer’s contributions to the field have been critical in establishing this link. We believe that this information is better suited to the discussion section given that enhanced cytokine production was observed in the results section and, as correctly asserted by the reviewer, only minor effects on direct cytotoxicity was observed. Taking on board the reviewers comments with regards to additional references we have added the following text to the discussion.

Line 430: “This is likely related to the capacity of IFN γ and TNF to evoke ‘bystander’ activity on tumor cells not directly targeted by the CAR T cells³³, modulation of other immune cells within the tumor microenvironment and/or its propensity to modulate the tumor stroma and inhibit neovascularization as previously observed in the context of A_{2A}R deficient T cells⁷”

We have also added the recommended discussion with regards to A_{2B}R vs. A_{2A}R targeting on line 379:

“Our results are consistent with previous studies indicating that A_{2A}R is the major target for adenosine on T cells and that other adenosine receptors such as the A_{2B}R do not compensate for the loss of A_{2A}R⁵². This may be due, in part, to the observation that A_{2A}R is required for expression of A_{2B}R on the cell surface⁵³. Nevertheless, these concepts support our observation that targeting the A_{2A}R with CRISPR/Cas9 is sufficient to attenuate adenosine mediated suppression of CAR T cells.

2nd Set of Suggestions:

My advice is to revise the beginning of the Introduction with which is something like:

1 =CAR-T cell were shown to be effective in rejecting tumors in lymphoma patients, but CAR-T failed to reject solid tumors, prompting US-i.e. Beavis/Darcy- to test and provide the proof of principle for the therapeutic promise of the blockade of A_{2A}R in preclinical studies.

Therefore the ref#1 in this paper should be ref on the earlier CAR-T cells paper from these Authors as the proof of principle for this specific paper.

=Beavis, P.A. et al. Targeting the adenosine 2A receptor enhances 827 chimeric antigen receptor T cell efficacy.
J Clin Invest 127, 929-941 (2017).

=Then I suggest to proceed with providing the first and strongest justifications to genetically eliminate the cAMP-elevating A2AR in adoptively transferred CTL as is the Aim of submitted paper.

- Therefore the ref#2,3 should be on Ohta et al Nature 2001 and 2006 PNAS papers which created this Anti-A2-adenosinergic field of cancer immunology.

=Ref 2 Ohta A, Sitkovsky M Role of G-protein-coupled adenosine receptors in downregulation of inflammation and protection from tissue damage. Nature. 2001 Dec 20-27;414(6866):916-20.

Ref 3 Ohta A, Gorelik E, Prasad SJ, Ronchese F, Lukashev D, Wong MK, Huang X, Caldwell S, Liu K, Smith P, Chen JF, Jackson EK, Apasov S, Abrams S, Sitkovsky M. A2A adenosine receptor protects tumors from antitumor T cells. Proc Natl Acad Sci U S A. 2006 Aug 29;103(35):13132-7.

Response: We thank the reviewer for their suggestions. As mentioned above our introduction aims to provide context to the reader from the perspective of both the CAR T cell and adenosine fields. Therefore our first references are relevant to the CAR T cell field. However, we have restructured the introduction to incorporate references Ohta et al. 2001 and Ohta et al. 2006 earlier and more prominently as is warranted by their influence on this work.

3rd Set of Suggestions:

To improve the Discussion by explaining the possible limitations of the proposed approach and what to be alert about in terms of safety and efficacy:

1. To provide references on the most important Reviews-Opinions, not just Reviews with collection of factoids???

I recommend the

=Vijayan D, Young A, Teng MWL, Smyth MJ. Targeting immunosuppressive adenosine in cancer. Nat Rev Cancer. 2017 Dec;17(12):709-724. Review.

=Cancer Discov. 2020 Jan;10(1):16-19.Lessons from the A2A Adenosine Receptor Antagonist-Enabled Tumor Regression and Survival in Patients with Treatment-Refractory Renal Cell Cancer. Sitkovsky MV.

= The ???Hostile TME???.??? In Cancer Immunology Research by Sitkovsy and Hatfield???

Response: We thank the reviewer for their suggestion and have referenced the recommended reviews/commentaries.

Important Safety Warning:

Authors should discuss the issue of autoimmunity due to the A2AR deletion in T cells, which was first actually noticed and emphasized in PNAS 2006 paper including the possible solution by restarting this A2AR pathway???

From the Abstract of PNAS 2006 paper:

- The observations of autoimmunity during melanoma rejection in A2AR-deficient mice suggest that A2AR in T cells is also important in preventing autoimmunity. Thus, although using the hypoxiaadenosineA2AR pathway inhibitors may improve antitumor immunity, the recruitment of this pathway by selective drugs is expected to attenuate the autoimmune tissue damage.

Response: We do not believe that autoimmunity is a significant concern in the context of CAR T cells since this has never been reported in any clinical trial for solid tumors with CAR T cells. Moreover our experimental data specifically addresses the possible concern of autoimmune related toxicities and observed no evidence for this. However to address this concern we have incorporated the following text on line 456:

“From a safety perspective, although there may be potential concerns with permanent A_{2A}R deletion and excessive immune responses directed against the host, we observed that A_{2A}R-edited CAR T cells were safe and well tolerated as shown by unaltered homeostatic concentrations of enzymes that are indicative of liver or kidney toxicities and histology of tissue sections of mice that had undergone therapy. Indeed, targeting the A_{2A}R is likely to have a favorable safety profile relative to other immunosuppressive pathways given that the hypoxia-adenosine axis is more prominent in the tumor microenvironment.”

3-2

To discuss that even the A2AR deficient CARs could be STILL inhibited by other immunosuppressive molecules

-like TGF-beta- and T regs and MDSC in the most resistant to inhibition hypoxia-HIF and extAdo??A2AR??cAMP-rich TMEs.

This is due to the ???hypoxia-HIF-1alpha-HRE mediated and extAdo??A2AR-cAMP-mediated immunosuppressive transcription???

-which was the conceptually novel concept when it was put forward 11 years ago by Sitkovsky MV in T regulatory cells: hypoxia-adenosinergic suppression and re-direction of the immune response. Trends Immunol.Cell, 2009 Mar;30(3):102-8.

Response: We thank the reviewer for the suggestion. To address the topic of immunosuppression by other pathways we have added the following text to the end of the discussion.

“Given that A_{2A}R-edited CAR T cells will remain subject to these other immunosuppressive pathways, we believe that in future studies it will be of interest to investigate targeting multiple suppressive genes using CRISPR/Cas9 to further protect CAR T cells against tumor-induced immunosuppression.”

3-3

Authors should address the troubling possibility of compensation for immunosuppressive A2AR by also immunosuppressive , cAMP-elevating A2BR in their CAR-T cells.

If likely, this would defeat the purpose of depleting the CAR-T-cells from the A2AR.

Authors may stronger justify their approach by referring to the experimental evidence that the immunosuppressive and cAMP-elevating A2BR adenosine receptor does not compensate for A2AR:

Ref Lukashev DE, Smith PT, Caldwell CC, Ohta A, Apasov SG, Sitkovsky MV Analysis of A2a receptor-deficient mice reveals no significant compensatory increases in the expression of A2b, A1, and A3 adenosine receptors in lymphoid organs.
Biochem Pharmacol. 2003 Jun 15;65(12):2081-90.

It was reported in this paper, that

???"These data suggest that regulation of the expression of A2b, A1, and A3 receptors is not affected significantly by the absence of A2a receptors and may provide further explanation of earlier in vivo observations of increased tissue damage and of longer persistence of proinflammatory cytokines in animals with inactivated A2a receptors???"

3-4

Mention that the likely reason that the immunosuppressive A2BR can not compensate for A2AR is because the A2AR is required for the A2BR expression on the surface of lymphocytes.

???" =Ref on Moriyama K, Sitkovsky MV Adenosine A2A receptor is involved in cell surface expression of A2B receptor.
J Biol Chem. 2010 Dec 10;285(50):39271-88.

-Thus, Authors should emphasize that such highly undesirable compensatory upregulation of immunosuppressive A2BR in their CAR-T with the deletion of A2AR.

Response: We thank the reviewer for the suggestion. We have added the recommended discussion with regards to A_{2B}R vs. A_{2A}R targeting on line 379:

"Our results are consistent with previous studies indicating that A_{2A}R is the major target for adenosine on T cells and that other adenosine receptors such as the A_{2B}R do not compensate for the loss of A_{2A}R⁵². This may be due, in part, to the observation that A_{2A}R is required for expression of A_{2B}R on the cell surface⁵³. Nevertheless, these concepts support our observation that targeting the A_{2A}R with CRISPR/Cas9 is sufficient to attenuate adenosine mediated suppression of CAR T cells."

4th Set of Suggestions:

What is the roadmap to future improvement?

Authors may offer the therapeutic prevention of the still strong inhibition of A2AR-deficient CAR-T cells in the hypoxic and adenosine-rich TME by OTHER immunosuppressors that are triggered by hypoxia-HIF-1 and adenosine-A2AR-cAMP-mediated immunosuppressive transcription, that could be much more influential in ???old??? human tumors than in 2-3 week old mouse tumors

Authors can find these therapeutic prevention measures recently described in the invited Opinions and Reviews below:

1. Sitkovsky MV.

Lessons from the A2A Adenosine Receptor Antagonist-Enabled Tumor Regression and Survival in Patients with Treatment-Refractory Renal Cell Cancer.

Cancer Discov. 2020 Jan;10(1):16-19.

and these measures include the STOP of the Hypoxia-A2A adenosinergic pathway upstream by eliminating the TME hypoxia

2. Hatfield SM, Sitkovsky MV.

Antihypoxic oxygenation agents with respiratory hyperoxia to improve cancer immunotherapy.

J Clin Invest. 2020 Nov 2;130(11):5629-5637.

Response: We thank the reviewer for the suggestion. We have added the recommended discussion with regards to targeting tissue hypoxia on line 454:

“However, it may be of interest to combine this approach with other strategies designed to target tissue hypoxia such as supplemental oxygenation or hypoxia sensitive pro-drugs^{59, 60}”.

5th Set of Suggestions:

5-1

Authors should improve the statement below by referencing not only their own 2015/2017 studies but also much earlier and careful 1997 paper???

???A major immunosuppressive phenotype mediated by A2AR activation is impaired cytokine production by T cells whilst cytotoxic activity is less affected^{16, 29}.

Ref 16= Beavis, P.A. et al. Targeting the adenosine 2A receptor enhances 827 chimeric antigen receptor T

cell efficacy. J Clin Invest 127, 929-941 (2017).

Ref 29 Beavis, P.A. et al. Adenosine Receptor 2A Blockade Increases the Efficacy of Anti-PD-1 868 through Enhanced Antitumor T-cell Responses. Cancer Immunol Res 3, 506-517 (2015).

-The correct reference that started these comparisons is in

a.Koshiba M, Kojima H, Huang S, Apasov S, Sitkovsky MV..Memory of extracellular adenosine A2A purinergic receptor-mediated signaling in murine T cells. J Biol Chem. 1997 Oct 10;272(41):25881-9. It was shown that extracellular Ado (extAdo) suppresses all tested T cell receptor (TCR)-triggered effector functions of T lymphocytes including the stronger inhibition of the TCR-triggered FasL mRNA up-regulation in cytotoxic T lymphocytes .

b.

Cutting edge: Physiologic attenuation of proinflammatory transcription by the Gs protein-coupled A2A adenosine receptor in vivo.

Lukashev D, Ohta A, Apasov S, Chen JF, Sitkovsky M. J Immunol. 2004 Jul 1;173(1):21-4. doi:

c. And very relevant reference that was first to show in vivo in tumor rejections assays that A2AR/A2BR Antagonists May Deinhbit the Production of IFN-?? by Antitumor T Cells in the Tumor Microenvironment???

Ohta A, Gorelik E, Prasad SJ, Ronchese F, Lukashev D, Wong MK, Huang X, Caldwell S, Liu K, Smith P, Chen JF, Jackson EK, Apasov S, Abrams S, Sitkovsky M.

A2A adenosine receptor protects tumors from antitumor T cells. Proc Natl Acad Sci U S A. 2006 Aug 29;103(35):13132-7.

Response: We thank the reviewer for the suggestion. We have added the recommended references.

5-2

Authors must change this only partially correct statement:

As is , Authors write:

???'Two previous studies, including our own, have characterized the effect of shRNA in modulating A2AR mediated suppression

References =16, 42

but these studies were limited to in vitro assessment of cytokine production and killing capacity.

16-Beavis, P.A. et al. Targeting the adenosine 2A receptor enhances chimeric antigen receptor T cell efficacy. J Clin Invest 127, 929-941 (2017)

42-Masoumi, E. et al. Genetic and pharmacological targeting of A2a receptor improves function of anti-mesothelin CAR T cells. J Exp Clin Cancer Res 39, 49 (2020).

In fact, the first, original justifications to genetically eliminate A2AR in adoptively transferred CTL that is the main aim of submitted study were presented in Ohta at al.PNAS

Therefore, describing the effect of shRNA Authors should reference the PNAS 2006 paper by Ohta at all:

In that paper the genetic, siRNA targeted knock-down of A2AR by pretreatment of anti-tumor T cells before their adoptive transfer into tumor-growing mice was shown to result in better anti-tumor immunity in vivo and that was interpreted as one of the several presented lines of evidence, that effects of A2AR deletion are T cell-autonomous.

Ref Ohta A, Gorelik E, Prasad SJ, Ronchese F, Lukashev D, Wong MK, Huang X, Caldwell S, Liu K, Smith P, Chen JF, Jackson EK, Apasov S, Abrams S, Sitkovsky M. and

A2A adenosine receptor protects tumors from antitumor T cells. Proc Natl Acad Sci U S A. 2006 Aug 29;103(35):13132-7.

Response: We agree with the reviewer that this paper (Ohta et al. 2006) was important for showing the potential of A_{2A}R knockdown. Our sentence was intended to refer specifically to CAR T cells, and we have now amended this to reflect this.

5-3

Coming back to the i) importance of the evidence that effects of A_{2A}R deletion on improvement of tumor rejection are T-CELL-AUTONOMOUS.

This was prominently messaged in the Abstract PNAS 2006 by Ohta at all. and then in Cekic C, Linden J. paper ??? A_{2A} receptors intrinsically regulate CD8 T cells in their title Adenosine A_{2A} receptors intrinsically regulate CD8⁺ T cells in the tumor microenvironment. Cekic C, Linden J. Cancer Res. 2014 Dec 15;74(24):7239-49,

As a Reviewer with 38 years of publishing in top journals immunological studies of the T cell selection in thymus, TCR- activation, differentiation and effector functions, I must alert Authors about weaknesses in papers they refer to in order to protect and not to mislead the Reader.

This is why I recommend Authors to critically re-visit the paper from Cecic-Linden, before referencing it in the prestigious Nature Communications..

In my view this paper has several disqualifying mistakes in design and interpretations and in scholarship. I am very surprised that Authors of that paper succeeded in persuading their Reviewers and the Editor of Cancer Res. that their message is really novel (!?), so that they were even allowed to have it as title! - even that their paper was 8 years later after the PNAS 2006 paper where this same message was prominently displayed even in the Abstract.

I think they managed to persuade their Reviewers and the Editor of Cancer Res. that their message is really novel by employing a clever ???word play???, so that instead of ???T cell AUTONOMOUS??? wording that Ohta at all was prominently messaging 8 years earlier they employed the different, wording i.e. the synonymous ??? A_{2A} receptors INTRINSICALLY regulate CD8 T cells.

While this word play on itself is not sufficient to avoid referencing this paper, the poor experimental design and the over-interpretation of their observations are indeed, serious disqualifying problems.

This is why it is ???Poor experimental design??? that leads to artifacts; -Linden???'s paper was done with T cell specific A_{2A}R-knock-down mice, which are the ???immunological freaks of nature??? in terms of differentiation, and memory T cells due to the screwed up negative and positive selection in thymus of the A_{2A}R-/- thymocytes.

The TCR-repertoire and subsequent T cells memory ???signals??? are ALSO screwed up. -As a result, the behavior of the resulting peripheral T cells will be also abnormal and of relevance of only to this A_{2A}R-knock-down mice.

Indeed, memory T cells processes are heavily dependent on the TCR signaling???

There is only so much you can accept in terms of ???hints??? from such highly artificial mice without making special pharmacological controls or what Beavis and Darcy did in their papers.

I accept only the small part of observations trustworthy in Cecic-Linden paper and only because they are in agreement with earlier published controls by other good scientists.

This is why Authors of CAR-T paper do not need to ???validate??? in discussion chapter their strong and straightforward observations with the weak data in the -what I consider inferiorly designed and over-interpreted experiments and predictions.

##It must be noticed that the negative predictions of in Cecic-Linden paper about using A2AR blockade method, were NOT confirmed in subsequent clinical trials of A2AR blockade.

FYI,

according to the imaging radiology controls in humans , the blockade with A2AR blocker from Vernalis in Corvus trial is 100% of A2AR binding sites, so this is actually the ???Pharmacological KO??? of A2AR in humans.

Accordingly, the foundational Ohta at al, PNAS 2006 paper was both about i) A2AR KO and ii) about the pharmacological KO of A2AR in mouse WT anti-tumor T cells, that were not irreversibly screwed up.

Reviewer

Michail Sitkovsky

Response: We believe that the effect of A_{2A}R inhibition on CD8⁺ T cell differentiation is an interesting aspect that warrants further investigation. We note that the above study from Cecic-Linden did indeed use T cell specific A_{2A}R knockout mice, but these results were also confirmed using OT-I T cells derived from either A_{2A}R^{+/+} or A_{2A}R^{-/-} (globally deleted) mice, which would seemingly discount the theory that this result is entirely due to the use of T cell specific knockout mice. There is a clear link between enhancing T cell activation and promoting a terminally differentiated phenotype in the context of other immunosuppression pathways e.g. PD-1; PMID 26034050. Therefore one should not see a potential impact of A_{2A}R inhibition on T cell differentiation as a negative aspect necessarily. In our own hands we have data indicating that high doses of pharmacological antagonists of the A_{2A}R can mediate a similar effect as observed in the Cecic-Linden paper. However, we acknowledge this is a controversial area and we would need a more complete data set to make firm conclusions on this. Hence, we believe that a full discussion of this topic is beyond the scope of our already lengthy discussion and so we prefer to keep the text as it is in relation to this literature.